# DSI2I: Dense Style for Unpaired Exemplar-based Image-to-Image Translation

**Baran Ozaydin**  *baran.ozaydin@epfl.ch*
*School of Computer and Communication Sciences, EPFL, Switzerland*

**Tong Zhang**  *tong.zhang@epfl.ch*
*School of Computer and Communication Sciences, EPFL, Switzerland*

**Sabine Süsstrunk**  *sabine.susstrunk@epfl.ch*
*School of Computer and Communication Sciences, EPFL, Switzerland*

**Sabine Süsstrunk**  *sabine.susstrunk@epfl.ch*
*School of Computer and Communication Sciences, EPFL, Switzerland*

**Reviewed on OpenReview:** *https://openreview.net/forum?id=mrJi5kdKA4*

## Abstract

Unpaired exemplar-based image-to-image (UEI2I) translation aims to translate a source image to a target image domain with the style of a target image exemplar, without ground-truth input-translation pairs. Existing UEI2I methods represent style using one vector per image or rely on semantic supervision to define one style vector per object. Here, in contrast, we propose to represent style as a dense feature map, allowing for a finer-grained transfer to the source image without requiring any external semantic information. We then rely on perceptual and adversarial losses to disentangle our dense style and content representations. To stylize the source content with the exemplar style, we extract unsupervised cross-domain semantic correspondences and warp the exemplar style to the source content. We demonstrate the effectiveness of our method on four datasets using standard metrics together with a localized style metric we propose, which measures style similarity in a class-wise manner. Our results show that the translations produced by our approach are more diverse, preserve the source content better, and are closer to the exemplars when compared to the state-of-the-art methods. Project page: `https://github.com/IVRL/dsi2i`

## 1 Introduction

Unpaired image-to-image (UI2I) translation aims to translate a source image to a target image domain by training a deep network using images from the source and target domains without ground-truth input-translation pairs. In the exemplar-based scenario (UEI2I), an additional target image exemplar is provided as input so as to further guide the style translation. Ultimately, the resulting translation should 1) preserve the content/semantics of the source image; 2) convincingly seem to belong to the target domain; and 3) adopt the specific style of the target exemplar image.

Some existing UEI2I strategies Huang et al. (2018); Lee et al. (2018) encode the style of the exemplar using a global, image-level feature vector. While this has proven to be effective for relatively simple scenes, it leads to undesirable artifacts for complex, multi-object ones, as illustrated in Fig. 1, where appearance information of the dominating semantic regions, such as sky, unnaturally bleeds into other semantic areas, such as the road, trees and buildings. Other UEI2I methods Bhattacharjee et al. (2020); Jeong et al. (2021); Kim et al. (2022); Shen et al. (2019) address this by computing instance-wise or class-wise style representations. However, they require knowledge of the scene semantics, e.g., segmentation masks or bounding boxes during training, which limits their applicability.

Source          Baseline          **DSI2I**          Exemplar

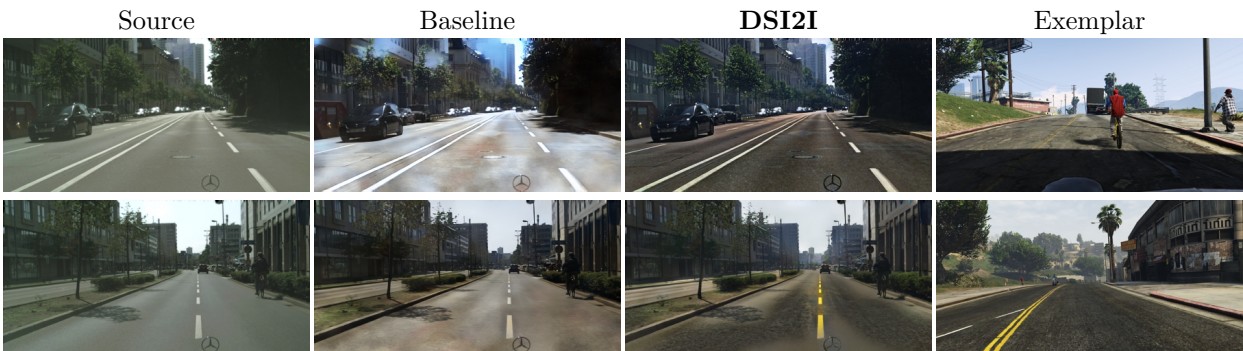

Figure 1: **Global style vs dense style representations.** The baseline method (MUNIT) Huang et al. (2018) represents the exemplar style with a single feature vector per image. As such, some appearance information from the exemplar bleeds into semantically-incorrect regions, giving, for example, an unnatural bluish taint to the road and the buildings in the second row, first image. By modeling style densely, our approach better respects the semantics when applying the style from the exemplar to the source content. Our method also has finer-grained control over style. The color of the road and center line in the third row reflect the exemplar appearance more accurately.

By contrast, we propose to model style densely. That is, we represent the style of an image with a feature tensor that has the same spatial resolution as the content one. The difficulty of having spatial information in style is that style information can more easily pollute the content one, and vice versa. To prevent this and encourage the disentanglement of style and content, we utilize perceptual and adversarial losses, which encourages the model to preserve the source content and semantics.

A dense style representation alone is not beneficial for UEI2I as the spatial arrangement of each dense style component is only applicable for its own image. Hence, we propose a cross-domain semantic correspondence module to spatially arrange/warp the dense style of the target image to the source content. To that end, we utilize the CLIP Radford et al. (2021) vision backbone as feature extractor and establish correspondences between the features of the source and target images using Optimal Transport Cuturi (2013); Liu et al. (2020).

As a consequence, and as shown in Fig. 1, our approach transfers the local style of the exemplar to the source content in a more natural manner than the global-style techniques. Yet, in contrast to Bhattacharjee et al. (2020); Jeong et al. (2021); Kim et al. (2022); Shen et al. (2019), we do not require semantic supervision during training, thanks to our dense modeling of style. To quantitatively evaluate the benefits of our approach, we introduce a metric that better reflects the stylistic similarity between the translations and the exemplars than the image-level metrics used in the literature such as FID Heusel et al. (2017), IS Salimans et al. (2016), and CIS Huang et al. (2018).

Our contributions can be summarized as follows:

- We propose a dense style representation for UEI2I. Our method retains the source content in the translation while providing finer-grain stylistic control.

- We show that adversarial and perceptual losses encourage the disentanglement of our dense style and content representations.

- We develop a cross-domain semantic correspondence module to warp the exemplar style to the source content.

- We propose a localized style metric to measure the stylistic accuracy of the translation.

Our experiments show both qualitatively and quantitatively the benefits of our method over global, image-level style representations. We will make our code publicly available upon acceptance.

| Method | Unpaired | Label Free | Multi -modal | Exemplar guided | Local style |
|---|:---:|:---:|:---:|:---:|:---:|
| MUNIT Huang et al. (2018) | ✓ | ✓ | ✓ | ✓ | ✗ |
| DRIT Lee et al. (2018) | ✓ | ✓ | ✓ | ✓ | ✗ |
| CUT Park et al. (2020) | ✓ | ✓ | ✗ | ✗ | ✗ |
| FSeSim Zheng et al. (2021) | ✓ | ✓ | ✓ | ✗ | ✗ |
| INIT Shen et al. (2019) | ✓ | ✗ | ✓ | ✓ | ✓ |
| DUNIT Bhattacharjee et al. (2020) | ✓ | ✗ | ✓ | ✓ | ✓ |
| MGUIT Jeong et al. (2021) | ✓ | ✗ | ✓ | ✓ | ✓ |
| CoCosNet Zhang et al. (2020) | ✗ | ✗ | ✓ | ✓ | ✓ |
| MCLNet Zhan et al. (2022b) | ✗ | ✗ | ✓ | ✓ | ✓ |
| MATEBIT Jiang et al. (2023) | ✗ | ✗ | ✓ | ✓ | ✓ |
| **DSI2I** | ✓ | ✓ | ✓ | ✓ | ✓ |

Table 1: **Comparison of I2I methods.** Unpaired methods do not require ground-truth translation pairs. Label free methods do not require object or segmentation annotations. Multimodal methods can produce multiple translations for one content. Exemplar guided methods can stylize the translations based on an exemplar image. The methods that represent style object-wise or densely have local style control.

## 2 Related Work

Our method primarily relates to three lines of research: Image-to-image (I2I) translation, Style Transfer, and Semantic Correspondence. Our main source of inspiration is I2I research as it deals with content preservation and domain fidelity. However, we borrow concepts from Style Transfer when it comes to adopting exemplar style and evaluating stylistic accuracy. Furthermore, our approach to swapping styles across semantically relevant parts of different images is related to semantic correspondences.

### 2.1 Image-to-image Translation

We focus the discussion of I2I methods on the unpaired scenario, as our method does not utilize paired data. CycleGAN Zhu et al. (2017) was the first work to address this by utilizing cycle consistency. Recent works Hu et al. (2022); Jung et al. (2022); Park et al. (2020); Zheng et al. (2021) lift the cycle consistency requirement and perform one-sided translation using contrastive losses and/or self-similarity between the source and the translation. Many I2I methods, however, are unimodal, in that they produce a single translation per input image, thus not reflecting the diversity of the target domain, especially in the presence of high within-domain variance. Although some works Jung et al. (2022); Zheng et al. (2021) extend this to multimodal outputs, they cannot adopt the style of a specific target exemplar, which is what we address.

Some effort has nonetheless been made to develop exemplar-guided I2I methods. For example, Huang et al. (2018); Lee et al. (2018) decompose the images into content and style components, and generate exemplar-based translations by merging the exemplar style with the content of the source image. However, these models define a single style representation for the whole image, which does not reflect the complexity of multi-object scenes. By contrast, Bhattacharjee et al. (2020); Jeong et al. (2021); Kim et al. (2022); Mo et al. (2018); Shen et al. (2019) reason about object instances for I2I translation. Their goal is thus similar to ours, but their style representations focus on foreground objects only, and they require object-level (pseudo) annotations during training. Moreover, these methods do not report how stylistically close their translations are to the exemplars. Here, we achieve dense style transfer for more categories without requiring annotations and show that our method generates translations closer to the exemplar style while having comparable domain fidelity with that of the state-of-the-art methods.

### 2.2 Style Transfer

Style transfer aims to bring the appearance of a content image closer to a target image. The seminal work of Gatys et al. Gatys et al. (2016) achieves so by matching the Gram matrices of the two images via image-

based optimization. Li et al. (2017b) provides an analytical solution to Gram matrix alignment, enabling arbitrary style transfer without image based optimization. Huang & Belongie (2017) only matches the diagonal of the Gram matrices by adjusting the channel means and standard deviations. Li et al. (2017a) shows that matching the Gram matrices minimizes the Maximum Mean Discrepancy between the two feature distributions. Inspired by this distribution interpretation, Kolkin et al. (2019) proposes to minimize a relaxed Earth Movers Distance between the two distributions, showing the effectiveness of Optimal Transport in style transfer. Zhang et al. (2019) defines multiple styles per image via GrabCut and exchanges styles between the local regions in two images. Kolkin et al. (2019); Zhang et al. (2019) are particularly relevant to our work as they account for the spatial aspect of style. Chiu & Gurari (2022); Li et al. (2018); Yoo et al. (2019) aim to achieve photorealistic stylization using a pre-trained VGG based autoencoder. Kim et al. (2020); Liu et al. (2021); Yang et al. (2022) model texture- and geometry-based style separately and learn to warp the texture-based style to the geometry of another image. However, the geometric warping module they rely on makes their methods only applicable to images depicting single objects. Our dense style representation and our evaluation metric are inspired by this research on style transfer. Unlike these works, our image-to-image translation method operates on complex scenes, deals with domain transfer and does not require image based optimization.

### 2.3 Semantic Correspondence

Semantic correspondence methods aim to find semantically related regions across two different images. This involves the challenging task of matching object parts and fine-grained keypoints. Early approaches Barnes et al. (2009); Liu et al. (2010) used hand-crafted features. These features, however, are not invariant to changes in illumination, appearance, and other low-level factors that do not affect semantics. Hence, they have limited ability to generalize across different scenes. Aberman et al. (2018); Liu et al. (2020); Min et al. (2019) use ImageNet Simonyan & Zisserman (2014) pre-trained features to address this issue and find correspondences between images containing similar objects. However, these methods do not generalize to finding accurate correspondences across images from different modalities/domains.

Semantic correspondences have been explored in the context of image to image translation as well. In particular, Zhan et al. (2021; 2022b); Zhang et al. (2020); Zhou et al. (2021); Zhan et al. (2022a) use cross-domain correspondences to guide paired exemplar-based I2I translation. These methods are applicable to a single dataset where the two paired domains consist of segmentation labels and corresponding images. Specifically, they aim to translate segmentation labels to real images. In this case, both the I2I and semantic correspondence tasks benefit from the paired data, i.e., semantic supervision. We also use cross-domain correspondences, but unlike these works, our method is 1) unpaired and unsupervised, i.e., the ground-truth translation is unknown; 2) unsupervised in terms of semantics, i.e., we do not use segmentation labels during training; 3) applicable to translation between two datasets from different domains.

## 3 Method

Let us now introduce our UEI2I approach using dense style representations. To this end, we first define the main architectural components of our model. It largely follows the architecture of Huang et al. (2018) and is depicted in Fig 2. Given two image domains $\mathbf{X}, \mathbf{Y} \subset \mathbb{R}^{3 \times H'W'}$, our model consists of two style encoders $E_X^s, E_Y^s : \mathbb{R}^{3 \times H'W'} \to \mathbb{R}^{S \times HW}$, two content encoders $E_X^c, E_Y^c : \mathbb{R}^{3 \times H'W'} \to \mathbb{R}^{C \times HW}$, two generators $G_X$, $G_Y : \mathbb{R}^{C \times HW} \times \mathbb{R}^{S \times HW} \to \mathbb{R}^{3 \times H'W'}$, and two patch discriminators $D_X, D_Y : \mathbb{R}^{3 \times H'W'} \to \mathbb{R}^{S \times H''W''}$.

The content and style representations are then defined as follows. The content of image $\mathbf{x}$ is computed as $\mathbf{C}_x := E_X^c(\mathbf{x})$, and its dense style as $\mathbf{S}_x^{dense} := E_X^s(\mathbf{x})$. Note that the latter departs from the definition of style in Huang et al. (2018); here, instead of a global style vector, we use a dense style map with spatial dimensions, which will let us transfer style in a finer-grained manner. Nevertheless, we also compute a global style for image $\mathbf{x}$ as $\mathbf{S}_x^{global} := Avg(\mathbf{S}_x^{dense})$, where $Avg$ denotes spatial averaging and repeating a vector across spatial dimensions. Furthermore, we define a mixed style $\mathbf{S}_x^{mix} := 0.5\mathbf{S}_x^{global} + 0.5\mathbf{S}_x^{dense}$. As will be shown later, this mixed style will allow us to preserve the content without sacrificing stylistic control.

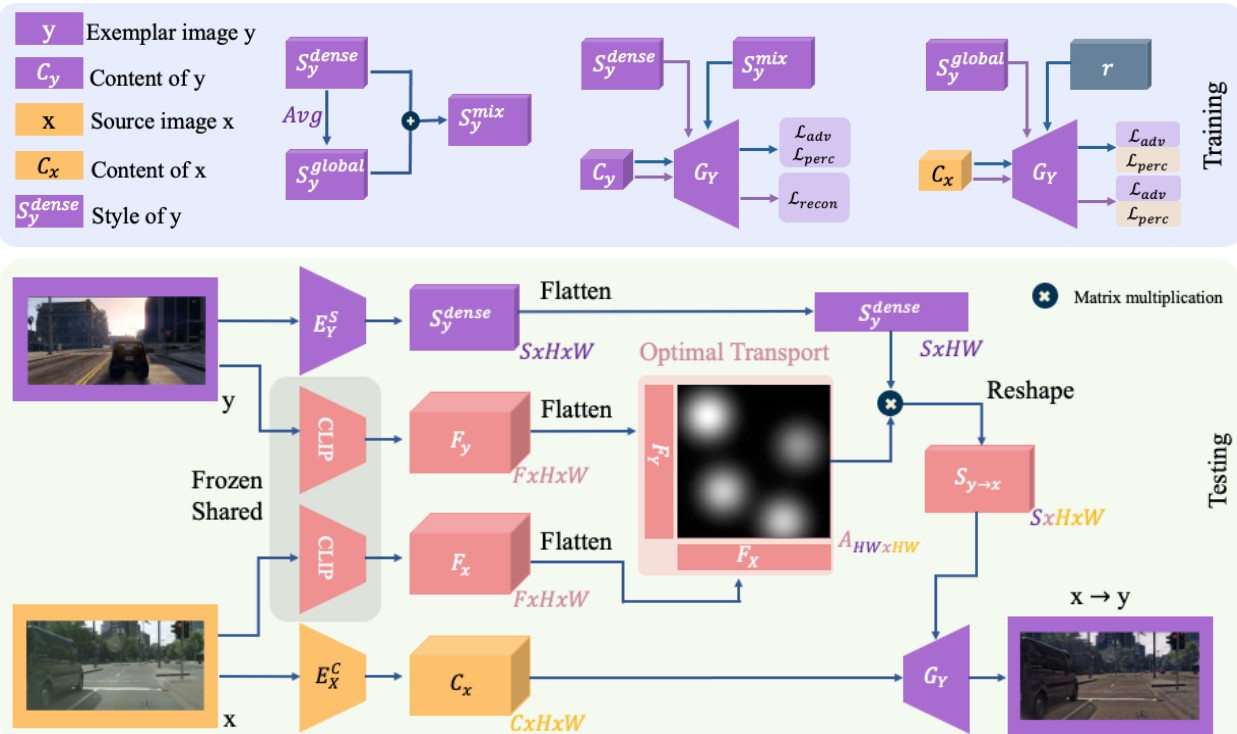

Figure 2: **Overview of method.** We represent style as a feature map with spatial dimensions and constrain it via adversarial and perceptual losses for disentanglement. Our method does not require any labels or paired images during training. In test time, we warp the style of the exemplar for the source content using semantic correspondence. At test time, we utilize the CLIP Radford et al. (2021) vision backbone to build semantic correspondences. See Section 3 for definitions and explanations.

In the remainder of this section, we first introduce our approach to learning meaningful dense style representations during training, shown in the top portion of Fig. 2. We then discuss how dense style is injected architecturally, and finally how to exchange the dense styles of the source and exemplar images at inference time, illustrated in the bottom portion of Fig. 2.

### 3.1 Learning Dense Style

We define style as low level attributes that do not affect the semantics of the image. These low level attributes can include lighting, color, appearance, and texture. We also believe that a change in style should not lead to an unrealistic image and should not modify the semantics of the scene. In this work, we argue that, based on this definition, style should be 1) represented densely to reflect finer grained stylistic attributes (stylistic accuracy); 2) constrained by an adversarial loss to encourage fidelity to the target domain (domain fidelity); 3) constrained by a perceptual loss to preserve semantics (content preservation).

To learn a dense style representation that accurately reflects the stylistic attributes of the exemplar, we utilize the $L_1$ reconstruction loss with $\mathbf{S}_y^{dense}$ to enable the flow of fine-grained dense information into the style representation. This is expressed as

$$L_{recon} = L_1(G_Y(\mathbf{C}_y, \mathbf{S}_y^{dense}), \mathbf{y}) . \tag{1}$$

Such an image reconstruction loss encourages the content and dense style representation to contain all the information in the input image. However, on its own, it does not prevent style from modeling content and leading to unrealistic or semantic changes when edited. To encourage a rich content representation that

preserves semantics, we use adversarial and perceptual losses with a random style vector $\mathbf{r} \sim \mathcal{N}(0, \mathbf{1}) \in \mathbb{R}^{S \times 1}$

$$L_{adv\_random} = L_{GAN}(G_Y(\mathbf{C}_x, \mathbf{r}), D_Y) \, , \tag{2}$$

$$L_{per\_random} = L_1(V(\mathbf{x}), V(G_Y(\mathbf{C}_x, \mathbf{r}))) \, , \tag{3}$$

These losses prevent our model from relying too much on dense style for reconstruction and translation by leading to a richer content representation.

Having a rich content does not prevent dense style from polluting it. To prevent style from modeling content, we constrain the dense style using the adversarial and perceptual losses

$$L_{adv\_global} = L_{GAN}(G_Y(\mathbf{C}_x, \mathbf{S}_y^{global}), D_Y) \, , \tag{4}$$

$$L_{per\_global} = L_1(V(\mathbf{x}), V(G_Y(\mathbf{C}_x, \mathbf{S}_y^{global}))) \, , \tag{5}$$

where $L_{GAN}$ denotes a standard adversarial loss, and $V$ represents the VGG16 backbone up to but excluding the Global Average Pooling layer. The adversarial loss above encourages the fidelity of the translations to the target domain Goodfellow et al. (2014); Zhu et al. (2017) whereas the perceptual losses help preserve the semantics Huang et al. (2018); Johnson et al. (2016); Zhu et al. (2017). While the global losses in Eqs. 4, 5 constrain dense style via the spatial averaging operation, there is no loss that involves $\mathbf{S}^{dense}$. Involving $\mathbf{S}^{dense}$ in the adversarial and perceptual losses tends to make the model learn to ignore the style representation, referred to as style collapse. Also, note that all the constraints in Eqs. 2, 3, 4, 5 use a spatially constant style representation. Hence, to involve a spatially varying style during the training and to avoid style collapse, we introduce two losses computed on the mixed style $\mathbf{S}_y^{mix}$, given by

$$L_{adv\_mix} = L_{GAN}(G_Y(\mathbf{C}_y, \mathbf{S}_y^{mix}), D_Y) \, , \tag{6}$$

$$L_{per\_mix} = L_1(V(\mathbf{y}), V(G_Y(\mathbf{C}_y, \mathbf{S}_y^{mix}))) \, , \tag{7}$$

### 3.2   Injecting Dense Style

Let us now describe how we inject a dense style map, $\mathbf{S}^{dense}$, in our framework to produce an image. Accurate stylization requires the removal of the existing style as an initial step Li et al. (2017b;a). Thus, for our dense style to be effective, we incorporate a dense normalization that first removes the style of each region. To this end, inspired by Li et al. (2019); Park et al. (2019); Zhu et al. (2020), we utilize a Positional Normalization Layer Li et al. (2019) followed by dense modulation. These operations are performed on the generator activations that produce the images.

Formally, let $\mathbf{P} \in \mathbb{R}^{C' \times HW}$ denote the generator activations, with $C'$ the number of channels. We compute the position-wise means and standard deviations of $\mathbf{P}$, $\mu, \sigma \in \mathbb{R}^{HW}$. We then replace the existing style by our dense one via the Dense Normalization (DNorm) function

$$F_{DNorm}(\mathbf{P}, \alpha, \beta) = \frac{\mathbf{P} - \mu}{\sigma}\beta + \alpha \, , \tag{8}$$

where the arithmetic operations are performed in an element-wise manner and by replicating $\mu$ and $\sigma$ $C'$ times to match the channel dimension of $\mathbf{P}$. The tensors $\alpha, \beta \in \mathbb{R}^{C' \times HW}$ are obtained by applying $1 \times 1$ convolutions to the dense style $\mathbf{S}^{dense}$.

Up to now, we have discussed how to inject dense style in an image and how to learn a meaningful dense style representation in the training stage. However, one problem remains unaddressed in the test stage: The dense style extracted from an image is only applicable to that same image because its spatial arrangement corresponds to that image. In this section, we therefore propose an approach to swapping dense style maps across two images from different domains.

Our approach is motivated by the intuition that style should be exchanged between semantically similar regions in both images. To achieve this, we leverage an auxiliary pre-trained network that generalizes well across various image modalities Radford et al. (2021). Specifically, we extract middle layer features $\mathbf{F}_x, \mathbf{F}_y \in \mathbb{R}^{F \times HW}$ by passing the source and exemplar images through the CLIP-RN50 backbone Radford

et al. (2021). We then compute the cosine similarity between these features, clipping the negative similarity values to zero. We denote this matrix as $\mathbf{Z}_{yx} \in \mathbb{R}^{HW \times HW}$ and use it to solve an optimal transport problem as described in Liu et al. (2020); Zhang et al. (2020). We construct our cost matrix $\mathbf{C}$ as

$$\mathbf{C} = \mathbf{1} - \mathbf{Z}_{yx} \text{ , with} \tag{9}$$

$$\mathbf{Z}_{yx} = \max(\cos(\mathbf{F}_y, \mathbf{F}_x), 0). \tag{10}$$

We then use Sinkhorn's algorithm Cuturi (2013) to compute a doubly stochastic optimal transportation matrix $\mathbf{A}_{yx} \in \mathbb{R}^{HW \times HW}$, which corresponds to solving

$$\mathbf{A}_{yx} = \arg \min_{\mathbf{A}} \langle \mathbf{A}, \mathbf{C} \rangle_F - \lambda h(\mathbf{A}) \tag{11}$$

$$\text{s.t} \quad \mathbf{A1}_{HW} = \mathbf{p}_y \text{ , } \mathbf{A}^T \mathbf{1}_{HW} = \mathbf{p}_x \tag{12}$$

where $h(\mathbf{A})$ denotes the entropy of $\mathbf{A}$ and $\lambda$ is the entropy regularization parameter. $\mathbf{p}_x$, $\mathbf{p}_y \in \mathbb{R}^{HW \times 1}$ constrain the row and column sums of $\mathbf{A}_{yx}$, which are chosen as uniform distributions (see the supplementary material for other choices). Optimal Transport returns a transportation plan $\mathbf{A}_{yx}$ to warp $\mathbf{S}_y^{dense}$ as

$$\mathbf{S}_{y \to x} = \mathbf{S}_y^{dense} \mathbf{A}_{yx} \text{ ,} \tag{13}$$

so that $\mathbf{S}_{y \to x}$ is semantically aligned with $\mathbf{x}$ instead of with $\mathbf{y}$. This plan transports style across semantically similar regions with the constraint that each region receives an equal mass. With this operation, each spatial element $\mathbf{S}_{y \to x}[h, w]$ can be seen as a weighted sum of spatial elements of $\mathbf{S}_y^{dense}[h', w']$ with the weights being proportional to the semantic similarity between $\mathbf{F}_x^{h,w}$ and $\mathbf{F}_y^{h'w'}$. Hence, we can trade the style across semantically similar regions.

Our semantic correspondence module can also be thought of as a cross attention mechanism across two images with the queries being $\mathbf{F}_x$, the keys $\mathbf{F}_y$ and the values $\mathbf{S}_y^{dense}$. Note also that global style transfer, as done in MUNIT Huang et al. (2018), is actually a special case of this formalism where $\mathbf{A}_{yx}$ is a constant uniform matrix.

### 3.3 Discussion on Losses and Components

**Discussion on the model components.** The semantic correspondence matrices $\mathbf{A}_{yx}$ built from CLIP Radford et al. (2021) features are 1) expensive in terms of computation and memory, and 2) noisy as each point corresponds to the others with some non-negative weight. For example, a self-correspondence matrix $\mathbf{A}_{xx}$(HWxHW) computed with CLIP Radford et al. (2021) would have large diagonal entries, positive but smaller off-diagonal entries for related semantic pixel pairs, and ideally zero off-diagonal entries for semantically unrelated pixel pairs. Instead of computing and storing these noisy and costly matrices with CLIP Radford et al. (2021) during training, we provide the losses with $\mathbf{S}^{mix}$ and $\mathbf{S}^{glb}$.

Our intuition for $\mathbf{S}^{mix}$ and $\mathbf{S}^{glb}$ is that these two style components replace noisy correspondence matrices of CLIP Radford et al. (2021) during training. $\mathbf{S}^{glb}$ is used to imitate cross-correspondences $\mathbf{A}_{yx}$ and can be seen as the output of a uniform, constant HWxHW correspondence matrix of 1/HW s (each content pixel corresponding to all the exemplar pixels equally) as shown in 3 parts a) and d); $\mathbf{S}^{dense}$ can be seen as the output of an identity self-correspondence matrix (each pixel corresponding only to itself) as shown in 3 parts b); and $\mathbf{S}^{mix}$ is the output of a noisy self-correspondence matrix, imitating $\mathbf{A}_{xx}$, with large diagonal entries and uniform non-diagonal entries (each pixel corresponds mainly to itself but also to all the others) as shown in 3 part c).

Additionally, randomly sampled style codes simulate a zero correspondence matrix, as shown in 3 part e), and enable our model to generalize to the cases where no style information (other than the random style vector) is available. This intuition is linked to the previous works on VAE Kingma & Welling (2013) and utilized for image translation in Liu et al. (2017); Huang et al. (2018).

Finally, these style components and analytical correspondence matrices enable our model to generalize to the HWxHW cross-correspondence matrices of CLIP Radford et al. (2021), without needing to use CLIP Radford et al. (2021) during training.

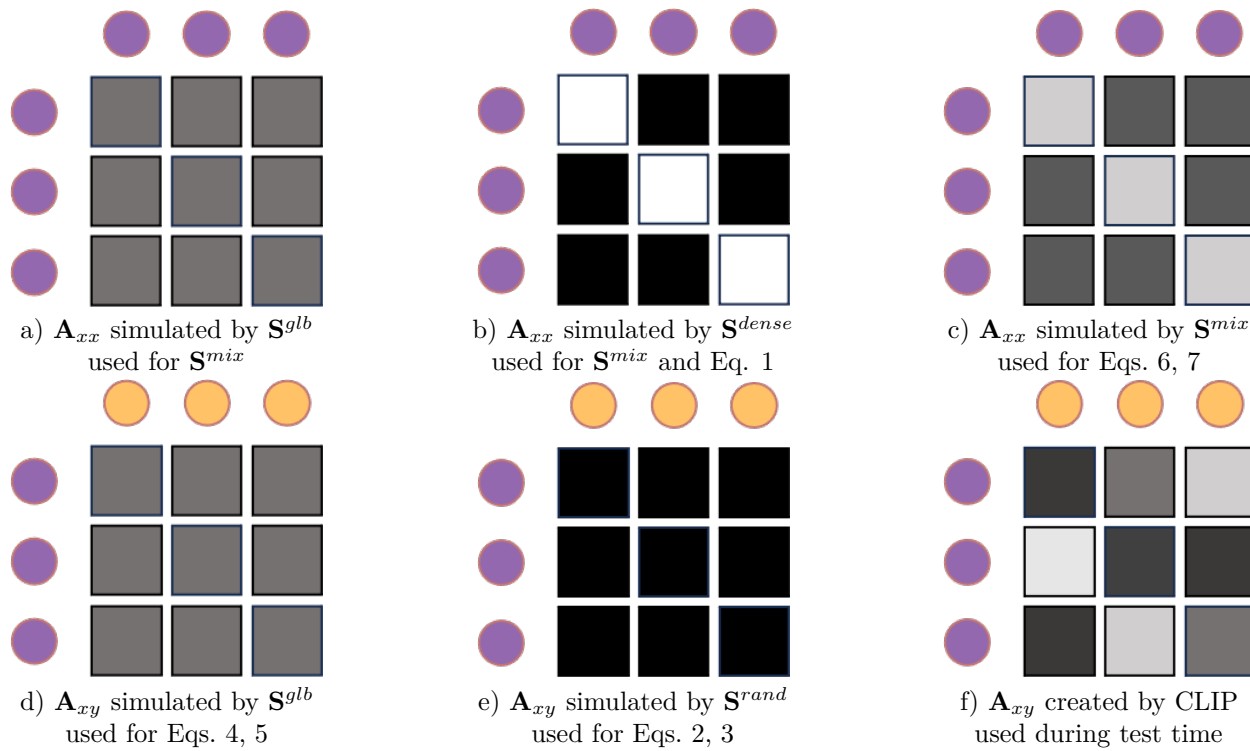

Figure 3: **Style components and correspondence matrices.** Example for the simulated and created correspondence matrices $\mathbf{A}_{xx}, \mathbf{A}_{xy} \in [0,1]^{3 \times 3}$. Top row includes the self-correspondence $\mathbf{A}_{xx}$ between three pixels from an image in the purple domain, whereas the bottom row displays cross-domain correspondence $\mathbf{A}_{xy}$ between an image from the purple domain and another image from the yellow domain. Using a)-e) during training enables our model to generalize to f) during test time.

**Discussion on the adversarial and perceptual losses.** Adversarial losses in our framework are mainly intended to produce translations that have high domain fidelity, whereas the perceptual losses are intended to preserve the content and semantics.

## 4 Experiments

### 4.1 Evaluation Metrics

In this UEI2I work, we have three goals and we evaluate these three goals with different metrics. To evaluate *stylistic accuracy*, we propose a novel metric to assess classwise stylistic distance that takes semantic information into account. To evaluate *domain fidelity* and how well the translations seem to belong to the target domain, we report the standard FID Heusel et al. (2017) between the translations and the targets. Lastly, to evaluate *content preservation*, we report segmentation accuracy with a segmentation model, DRN Yu et al. (2017), trained on the target domain and tested on the translations.

### 4.2 Classwise Stylistic Distance

Our local style metric, Classwise Stylistic Distance (CSD), computes the stylistic distance between the corresponding semantic classes in two images. We use VGG until its first pooling layer, denoted as $\hat{V}$, to extract features of size $\mathbb{R}^{V \times HW}$ from the input image $\mathbf{x}$, exemplar $\mathbf{y}$, and translation $\mathbf{x} \to \mathbf{y}$. Our metric uses binary segmentation masks $\mathbf{M}_x \in \mathbb{R}^{K \times HW}$ to compute the style similarity across corresponding classes. Using the mask for class $k$, $\mathbf{M}_x^k \in \mathbb{R}^{1 \times HW}$, we compute the Gram matrix $\mathbf{Q}_x^k$ of the VGG features for class

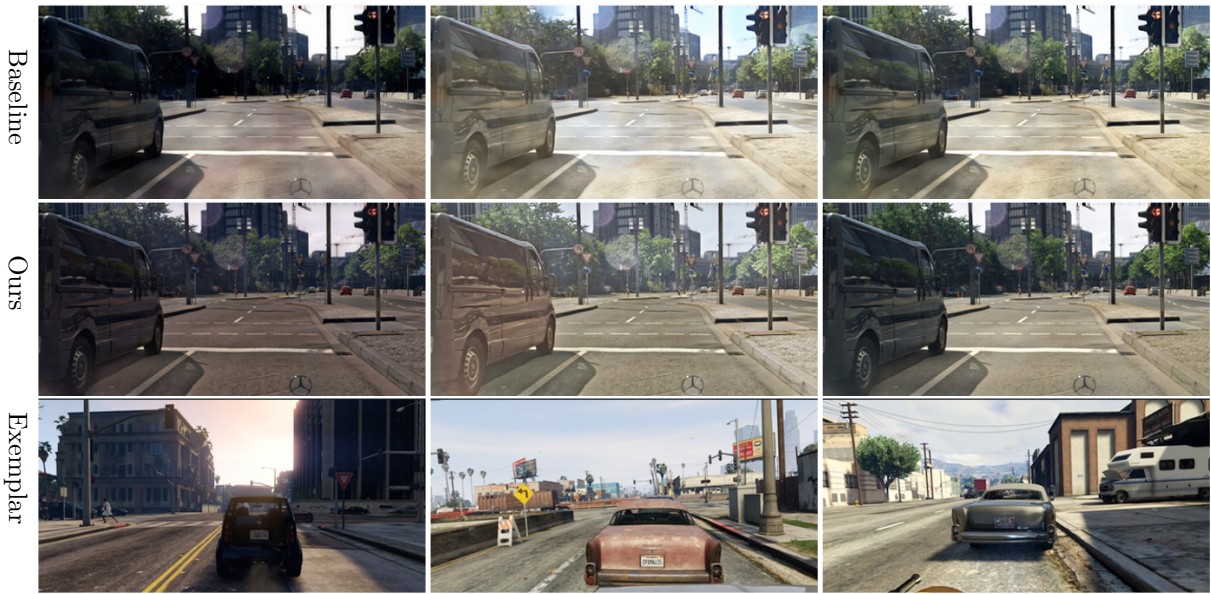

Figure 4: **Effect of the exemplar.** Our method can change the appearance of each semantic region differently, yet has realistic output. The colors of the road and car in the translations match the exemplar road and car styles better than the baseline (MUNIT) Huang et al. (2018) does. Content image can be seen in Figure 2

$k$ in image $\mathbf{x}$ as

$$\mathbf{Q}_x^k = \frac{1}{\sum_l \mathbf{M}_x^{k,l}} (\hat{V}(\mathbf{x}) \odot \mathbf{M}_x^k)(\hat{V}(\mathbf{x}) \odot \mathbf{M}_x^k)^T \ . \tag{14}$$

This operation is equivalent to treating each class as a separate image and computing their Gram matrices.

We then compute the distance between the Gram matrices of corresponding classes in two images, i.e.,

$$\mathbf{L}(\mathbf{x}, \mathbf{y}, k) = \|\mathbf{Q}_x^k - \mathbf{Q}_y^k\|_F^2 \ . \tag{15}$$

Note that $\mathbf{L}(\mathbf{x}, \mathbf{y}, k)$ denotes the Maximum Mean Discrepancy (MMD) between the features of the masked regions with a degree 2 polynomial kernel Li et al. (2017a). Since this distance is computed based on VGG features from an early layer, it implies a stylistic distance between the two images Gatys et al. (2016).

However, $\mathbf{L}(\mathbf{x}, \mathbf{y}, k)$ is not very informative as its scale is arbitrary and depends on the stylistic distance between the input image pair $\mathbf{x}, \mathbf{y}$. Hence, we propose a metric that takes $\mathbf{x}$, $\mathbf{y}$, and $\mathbf{x} \to \mathbf{y}$ at the same time for better interpretability. We express Classwise Stylistic Distance (CSD) as

$$\mathbf{H}(\mathbf{x}, \mathbf{y}, \mathbf{x} \to \mathbf{y}, k) = \frac{\mathbf{L}(\mathbf{x} \to \mathbf{y}, \mathbf{y}, k)}{\mathbf{L}(\mathbf{x}, \mathbf{y}, k)} \mathbb{1}_{\{\tilde{\mathbf{M}}_x^k > 0\}} \mathbb{1}_{\{\tilde{\mathbf{M}}_y^k > 0\}} \ , \tag{16}$$

where $\mathbb{1}_{\{\}}$ is the indicator function and $\tilde{\mathbf{M}}^k := \sum_l \mathbf{M}^{k,l}$.

Unlike $\mathbf{L}$, $\mathbf{H}$ is more interpretable because its value would be equal to one if the translation outputs the content image. In an ideal translation scenario, we would expect the feature distributions of the translation and exemplar to be close to each other Kolkin et al. (2019). Hence, we expect small values for more successful translations.

Note that Zhang et al. (2020) also proposes a metric to assess classwise stylistic similarity. Instead of the L2 distance between the classwise Gram matrices, it computes the cosine distance between the average features of corresponding regions in the exemplars and the translations. However, exemplar guided translation involves

three images; the source, the exemplar, and the translation. We believe that evaluating UEI2I should take all three images into account because stylistic distance between the source and exemplar affects the stylistic distance between the translation and exemplar, i.e. they are positively correlated. Our metric normalizes the stylistic distance between the exemplar and translation by the stylistic distance between the source and exemplar. By doing so, we obtain an interpretable value that shows which portion of the stylistic gap is closed for each translation, regardless of the initial style gap.

## 4.3 Implementation Details

We evaluate our method on real-to-synthetic and synthetic-to-real translations using the GTA Richter et al. (2016), Cityscapes Cordts et al. (2016), and KITTI Geiger et al. (2012) datasets. We use the code published by the baseline works Huang et al. (2018); Jeong et al. (2021); Lee et al. (2018); Park et al. (2020); Zheng et al. (2021). Images are resized to have a short side of 256. We borrow the hyperparameters from Huang et al. (2018) but we scale the adversarial losses by half since our method receives gradients from three adversarial losses for one source image. We do not change the hyperparameters for the perceptual losses. The entropy regularization term in Sinkhorn's algorithm in Eq. 12 is set to 0.05. During training, we crop the center 224x224 pixels of the images. During test time, we report single scale evaluation with the same resolution for all the metrics. We use a pre-trained DRNYu et al. (2017) to report the segmentation results.

We also evaluate our method on real-to-real translation using the sunny and night splits of the INIT Shen et al. (2019) dataset. We use the same setup as previous works and the results of the baselines are taken from the respective papers.

## 4.4 Results

| GTA → CS | car | sky | vege-tation | buil-ding | side-walk | road | Avg | KITTI → GTA | car | sky | vege-tation | buil-ding | side-walk | road | Avg |
|---|---|---|---|---|---|---|---|---|---|---|---|---|---|---|---|
| MUNIT Huang et al. (2018) | 0.43 | 0.78 | 0.21 | 0.28 | 0.13 | 0.06 | 0.32 | MUNIT Huang et al. (2018) | 0.46 | 0.17 | 0.59 | 0.39 | 0.64 | 0.53 | 0.46 |
| DRIT Lee et al. (2018) | 0.41 | 1.21 | 0.27 | 0.27 | 0.12 | 0.08 | 0.39 | DRIT Lee et al. (2018) | 0.52 | 0.22 | 0.61 | 0.44 | 0.85 | 0.55 | 0.53 |
| CUT Park et al. (2020) | 0.44 | 0.92 | 0.24 | 0.36 | 0.16 | 0.13 | 0.38 | CUT Park et al. (2020) | 0.53 | 0.21 | 0.63 | 0.47 | 0.87 | 0.76 | 0.57 |
| FSeSimZheng et al. (2021) | 0.40 | 0.96 | 0.25 | 0.38 | 0.15 | 0.13 | 0.38 | FSeSimZheng et al. (2021) | 0.50 | 0.25 | 0.76 | 0.49 | 0.88 | 0.83 | 0.61 |
| MGUIT Jeong et al. (2021) | 0.45 | 1.42 | 0.29 | 0.39 | 0.18 | 0.19 | 0.49 | MGUIT Jeong et al. (2021) | 0.40 | 0.21 | 0.74 | 0.47 | 0.84 | 0.55 | 0.53 |
| **DSI2I** | **0.29** | **0.22** | **0.16** | **0.26** | **0.08** | **0.03** | **0.17** | **DSI2I** | **0.29** | **0.08** | **0.42** | **0.34** | **0.59** | **0.23** | **0.32** |

Table 2: **Stylistic Accuracy.** Classwise Stylistic Distance between translation-exemplar pairs. Our translations match the classwise style of the exemplars better (lower is better).

**Stylistic Accuracy.** Firstly, we evaluate the stylistic distance between the exemplars and the translations using our metric CSD. We report this metric for the most frequent six classes of GTA Richter et al. (2016) and Cityscapes Cordts et al. (2016). The trend with other classes is similar and can be seen in our supplementary material. As shown in Table 2, our method outperforms the baselines in the synthetic-to-real and real-to-synthetic scenarios. Note that in the synthetic domains, stylistic diversity is overall higher because the images are more saturated. The results for translations in the opposite directions can be seen in our supplementary material. Our dense style and semantic correspondence modules bring style of corresponding classes closer to each other.

| | GTA → CS | | KITTI → GTA | | | sunny → night | | night → sunny | |
|---|---|---|---|---|---|---|---|---|---|
| Method | FID ↓ | Seg Acc ↑ | FID ↓ | Seg Acc ↑ | Method | CIS ↑ | IS ↑ | CIS ↑ | IS ↑ |
| MUNIT Huang et al. (2018) | 47.76 | 0.79 | 53.48 | 0.73 | MUNIT Huang et al. (2018) | 1.159 | 1.278 | 1.036 | 1.051 |
| DRIT Lee et al. (2018) | 42.93 | 0.70 | 52.12 | 0.62 | DRIT Lee et al. (2018) | 1.058 | 1.224 | 1.024 | 1.099 |
| CUT Park et al. (2020) | 49.82 | 0.65 | 62.30 | 0.59 | INIT Shen et al. (2019) | 1.060 | 1.118 | 1.045 | 1.080 |
| FSeSim Zheng et al. (2021) | 48.77 | 0.71 | 63.04 | 0.60 | DUNIT Bhattacharjee et al. (2020) | 1.166 | 1.259 | 1.083 | 1.108 |
| MGUIT Jeong et al. (2021) | 44.36 | 0.65 | 57.00 | 0.57 | MGUIT Jeong et al. (2021) | 1.176 | 1.271 | 1.115 | 1.130 |
| **DSI2I** | **42.61** | **0.82** | **48.30** | **0.75** | **DSI2I** | **1.204** | **1.283** | **1.138** | **1.149** |

Table 3: **Content preservation and domain fidelity.** Our method generates translations with high fidelity and preserves the content.

Table 4: **Diversity.** The translations produced by our method have higher diversity than those of the baselines.

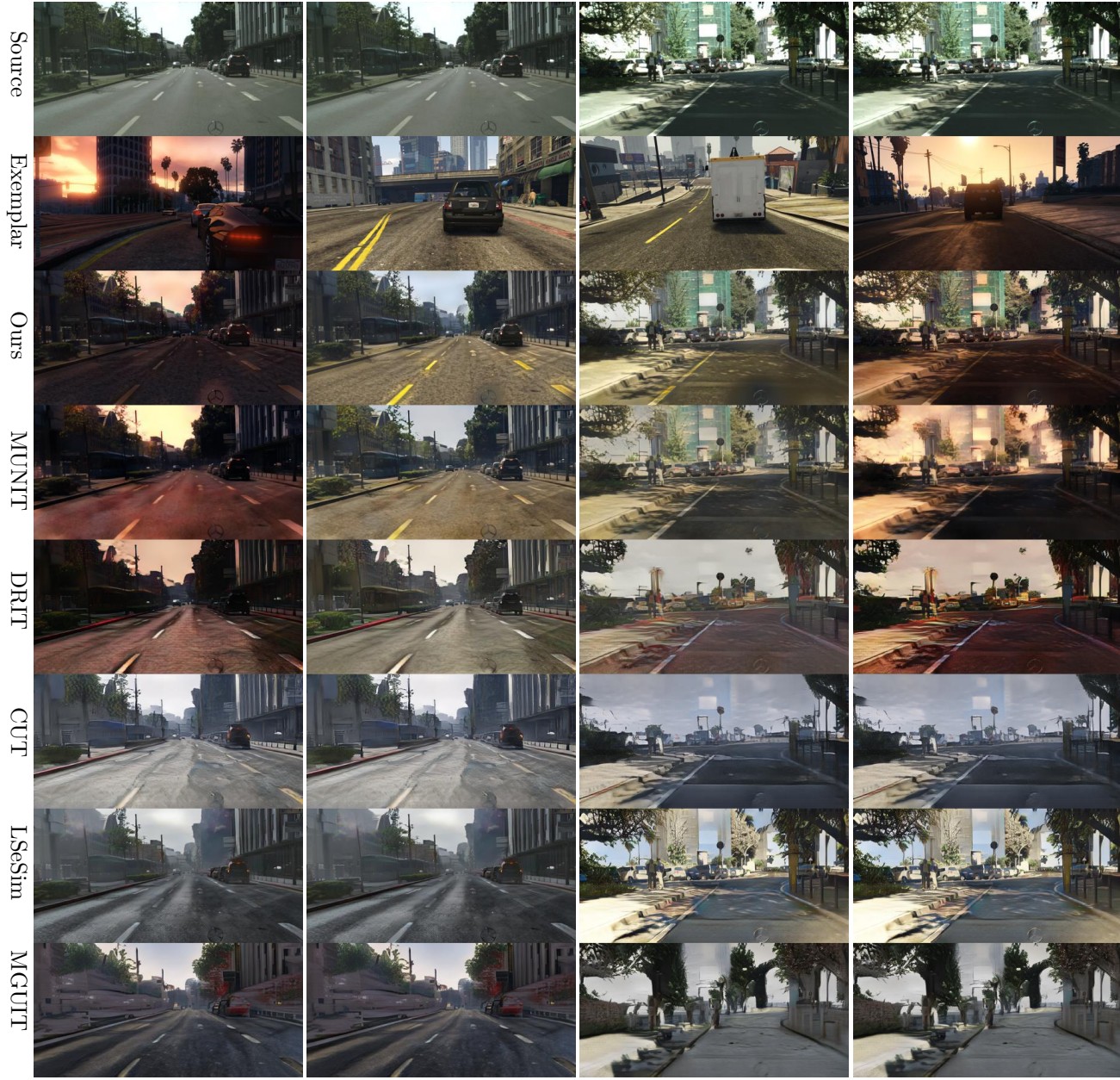

Figure 5: **Qualitative comparison with other methods.** CS → GTA translations. In the first column, our method disentangles the road from the sky and preserves the dark color for the road. In the second column, the appearance of the road and roadlines in our translation are closest to those in the exemplar. In the last two columns, our model preserves the semantics better, especially for tree and building classes.

**Domain Fidelity.** We then evaluate the domain fidelity of the translations using FID Heusel et al. (2017) in Table 3. Our method generates translations with high fidelity in the synthetic-to-real and real-to-synthetic scenarios, which pose large domain gaps.

**Content preservation.** We also evaluate how well our model preserves the content via segmentation accuracy in Table 3. Our method preserves content better than other I2I methods.

**Diversity.** Although our main goal is not diversity but stylistic accuracy, having a finer-grained dense style representation brings about diversity as a by product. We evaluate the diversity and quality of our

translations using the IS Salimans et al. (2016) and CIS Huang et al. (2018) metrics in real-to-real translation in Table 4. Our results are better than those reported in the baseline papers. Even though the baselines Bhattacharjee et al. (2020); Jeong et al. (2021); Shen et al. (2019) use object detection labels during training to guide style, we outperform them without using labels. Note that we do not use semantic correspondences, i.e., CLIP Radford et al. (2021), during training either. Hence, the performance increase is not due to dense semantic correspondences or the use of CLIP Radford et al. (2021) during training. Our dense style representation leads to greater stylistic control and diversity.

| Method | Test time label | FID ↓ | Styl. Dist. ↓ |
|---|---|---|---|
| CoCosNetv2 | GT Label | 46.32 | 0.34 |
| CoCosNetv2 | Pred Label | 51.32 | 0.37 |
| **DSI2I** | No Label | **45.12** | **0.32** |

| Method | FID ↓ | Styl. Dist. ↓ |
|---|---|---|
| CoCosNetv2 Zhou et al. (2021) | 51.32 | 0.37 |
| MCLNet Zhan et al. (2022b) | 50.42 | 0.38 |
| MATEBIT Jiang et al. (2023) | 49.25 | 0.36 |
| **DSI2I** | **45.12** | **0.32** |

Table 5: **Quantitative Comparison with CoCos-Netv2 Zhou et al. (2021).** Our method (without train-test time labels) outperform CoCosNetv2 (with train-test time labels). When the ground truth labels are replaced with the predicted labels (%95 accurate) CoCosNetv2 performance drops drastically.

Table 6: **Quantitative Comparison with Semantic Image Synthesis Methods** Our method outperforms the image synthesis baselines that use predicted labels.

**Comparison to exemplar guided semantic image synthesis.** Several works use semantic correspondence in I2I Zhan et al. (2021; 2022b); Zhang et al. (2020); Zhou et al. (2021); Zhan et al. (2022a); Jiang et al. (2023) to synthesize an image based on a given exemplar. As mentioned in Table 1, our method differs from this line of research in terms of training resources in three ways; 1) we do not require any semantic labels during training (Label Free), 2) our image translation task is not guided by ground truth translations during training (Unpaired), and additionally, 3) our method does not rely on highly similar exemplar-target pairs within the same domain.

To demonstrate the effectiveness of the unsupervised aspect of our method, we provide comparisons with the exemplar based image synthesis works. To that end, we train CoCosNetv2 Zhou et al. (2021) on the GTA dataset using the GTA labels. We test them with GTA images as the exemplars by giving 1) ground-truth labels of a CS image, 2) segmentation predictions of a CS image (%95 accurate) as inputs. Our method outperforms CoCosnetv2 Zhou et al. (2021) that use labels both during training and test time. Our method also outperforms more recent works Zhan et al. (2022b); Jiang et al. (2023) even though we do not use any labels or pretrained segmentation models neither during training nor during test time as seen in Table 6.

**User study.** We conduct a user study on Amazon Mechanical Turk and ask the users which translation is closer to the exemplar in terms of classwise style, color and appearance. We show the users one target image, and translations (CS → GTA) from the six methods in Fig. 5. Out of 3003 votes, our method received the most votes (1062), see Table 7. MUNIT Huang et al. (2018) is the second best model with 860 votes. Our method brings the style of semantically relevant regions closer to each other and is preferred by humans.

## 4.5 Ablation Study

Our ablations in Table 8 show that the losses on $\mathbf{S}^{mix}$ and $\mathbf{S}^{glb}$ encourage our model to preserve content and generate high-quality translations. The effects of adversarial and perceptual losses are shown in Table 9. Additional analysis on the model components can be found in the Appendix in Tables 12 13 14.

Tables 8 and 9 include ablations on GTA -> CS for our losses and style components (w/o $\mathbf{S}^{glb}$ is equivalent to w/o $L_{adv\_glb}$, $L_{perc\_glb}$; w/o $\mathbf{S}^{mix}$ is equivalent to w/o $L_{adv\_mix}$, $L_{perc\_mix}$). The adversarial losses are mainly helpful for domain fidelity (Table 9, FID column). The perceptual losses are mainly beneficial for content preservation (Table 9, Seg. Acc. column). $\mathbf{S}^{mix}$ and $\mathbf{S}^{glb}$ provide analytical noisy correspondences during training time and lead to better FID and Seg. Acc. in Table 8 during test time, when CLIP correspondences are used with OT. Altogether, our ablations in Tables 9 8 12 13 14 show that the adversarial

| Source | Ours | CoCosNetv2 | Exemplar |

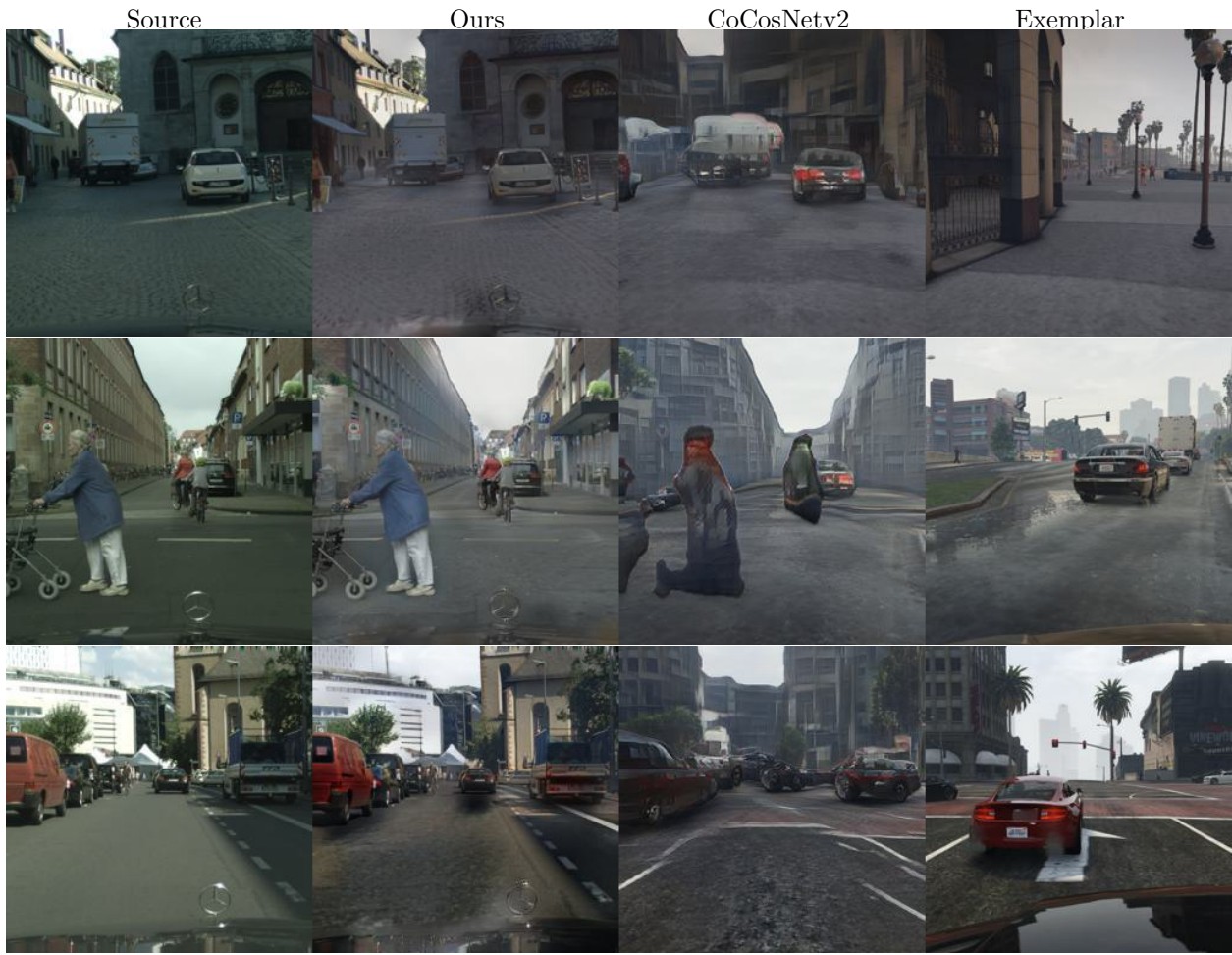

Figure 6: **Qualitative results from Table 5.** CS → GTA. CoCosNetv2 fails when Source and Exemplar images are from different domains and have uncommon classes. The human in the 2nd row, the car in the 1st and 3rd rows and the buildings in all rows are preserved better with our method. Our translations are more realistic and better represent the source content.

| Method | **DSI2I** | MUNIT | DRIT | CUT | FSeSim | MGUIT |
|--------|-----------|-------|------|-----|--------|-------|
| Ratio | **35%** | 28% | 18% | 5% | 7% | 5% |

Table 7: User study on similarity of translations with exemplars.

and perceptual losses on $\mathbf{S}^{glb}$ and $\mathbf{S}^{mix}$ are useful in terms domain fidelity (FID), content preservation (Seg. Acc.), and stylistic accuracy (Styl. Dis.).

## 5 Limitations

The main advantage of our method compared to the baselines is the dense modeling of style. Hence, our method loses its advantage for simple scenes with fewer objects where dense style is not necessary.

| GTA → CS | FID ↓ | Seg Acc ↑ |
|---|---|---|
| DSI2I | **42.61** | **0.82** |
| DSI2I w/o $\mathbf{S}^{glb}$ | 43.52 | 0.80 |
| DSI2I w/o $\mathbf{S}^{mix}$ | 45.64 | 0.78 |
| DSI2I w/o $\mathbf{S}^{mix}$, $\mathbf{S}^{glb}$ | 50.63 | 0.72 |

Table 8: Ablation study on $\mathbf{S}^{mix}$ and $\mathbf{S}^{glb}$. Our method benefits from both.

| GTA → CS | FID ↓ | Seg Acc ↑ |
|---|---|---|
| DSI2I | **42.61** | **0.82** |
| DSI2I w/o $L_{adv*}$ | 48.30 | 0.82 |
| DSI2I w/o $L_{perc*}$ | 42.96 | 0.73 |
| DSI2I w/o $L_{adv*}, L_{perc*}$ | 50.63 | 0.72 |

Table 9: Ablation study on adversarial and perceptual losses with $\mathbf{S}^{mix}$ and $\mathbf{S}^{glb}$. Adversarial loss encourages domain fidelity whereas perceptual loss helps preserve the content.

## 6 Conclusion

We present a framework for UEI2I that densely represents style and show how such a dense style representation can be learned and exchanged across images. This formalism allows local stylistic changes across semantic regions, while not requiring any labels. We demonstrate the effectiveness of our dense style representation in the synthetic-to-real, real-to-synthetic and real-to-real scenarios by showing that our translations match the style of the exemplar better, are more diverse, better preserve the content, and have high fidelity.

**Acknowledgements.** This work was supported by the Swiss National Science Foundation via the Sinergia grant CRSII5-180359. We also thank Ehsan Pajouheshgar for valuable discussions and contributions.

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

# A    Results on Unlabeled Datasets

We evaluated our method on datasets which have ground truth segmentation labels. The reason behind this is that, even though our method is applicable to datasets without labels, the metrics we care about (Seg. Acc. and Styl. Dist.) rely on ground-truth segmentation labels. KITTI, GTA and Cityscapes satisfy this label requirement for quantitative evaluation. Furthermore, some of the baselines, CoCosNetv2, MCLNet, MATEBIT and MGUIT, strictly rely on semantic segmentation labels for training, which makes them inapplicable to unlabeled datasets.

Our method, however, is applicable to scenes without semantic labels. Hence, as requested by the reviewers, we provide results on the summer2winter and monet2photo datasets Zhu et al. (2017), in both directions. Our qualitative results show that dense modeling of style enables more accurate transportation of style between semantically relevant regions.

Also, we would like to mention that, to our knowledge, our method is the first GAN-based I2I method to model style densely and exchange it accurately across semantic regions, in unlabeled datasets.

# B    Limitations

Our method is effective at preserving the content for a semantically distinct image pair from two semantically related datasets. The second row of Figure 5 is a good example, where our method preserves the content and styles of the pedestrian, bike and rider classes even though there are no such classes in the exemplar. Another example is provided in monet2photo in Figure 6, where the yellow leaf stylizes the vegetation in the ground (a semantically relevant but distinct class) but the other semantic classes are less affected and retain their style. However, our method is not effective for translation between semantically distinct dataset pairs. For example, in horse2zebra dataset, where the image translation requires semantic changes, our method often fails to add the stripes to the horses animals. In Figure 8, we show a cherry-picked result in the first row and another example that reflects the general performance of our method in the second row. The limitation might be partly due to the perceptual loss with VGG, which is too conservative for the horze2zebra task.

In our work, the attributes to be swapped are matched via CLIP-based correspondence whereas the attributes to be preserved are constrained via VGG-based perceptual loss. The choice of the VGG backbone reflects what kind of content we aim to preserve whereas the choice of the CLIP backbones reflects among which regions we aim to exchange the dense style. Hence, for the applications whose style/correspondence/content definitions differ from ours, a possible solution could be to experiment with other backbones instead of CLIP and VGG. An example for horse2zebra dataset would be to use a background focused perceptual loss instead of VGG-based perceptual loss and an animal-part focused correspondence backbone instead of the CLIP based correspondence. One reference for such a solution could be AttentionGAN Chen et al. (2018).

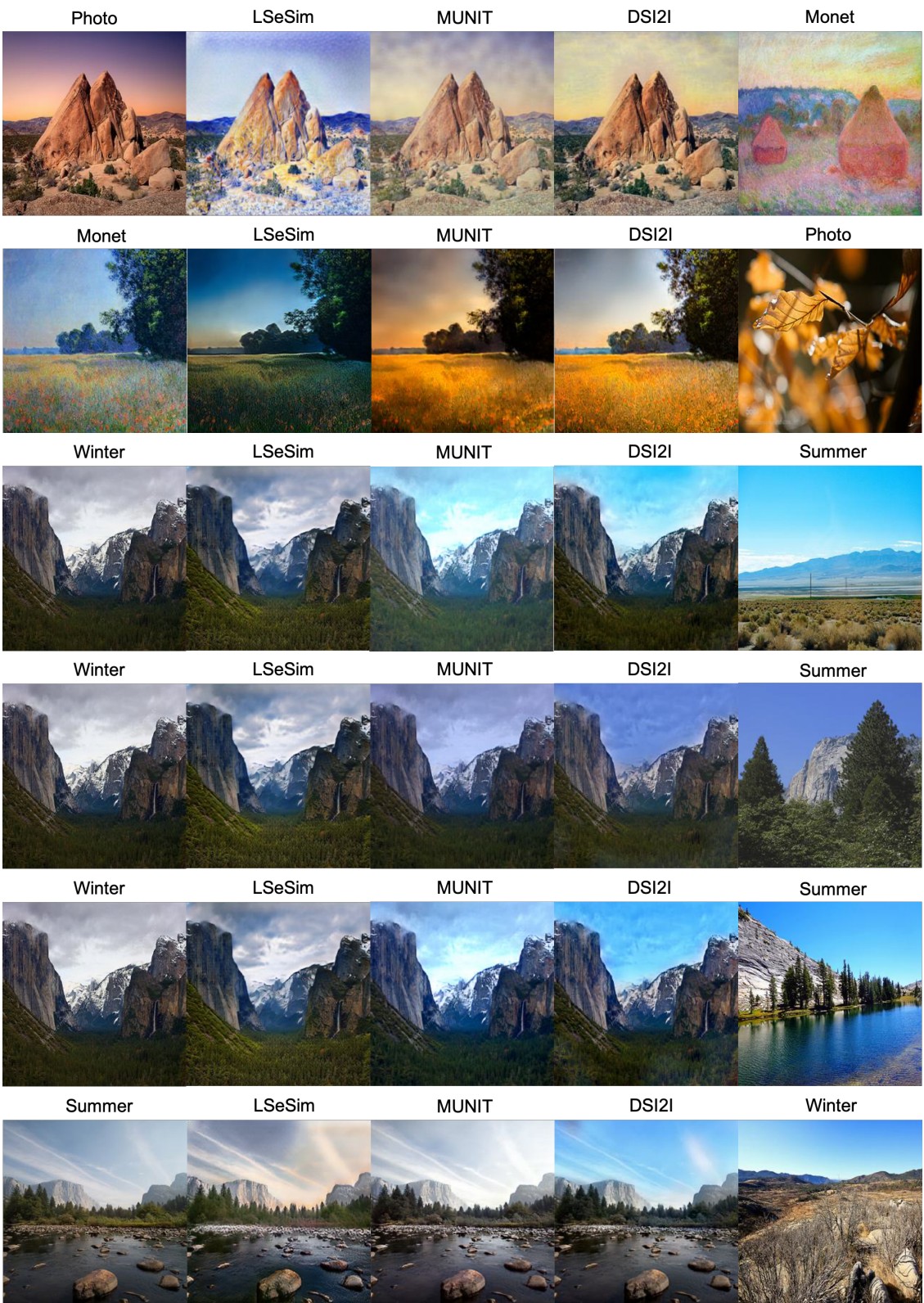

Figure 7: **Applicability of our method.** DSI2I is effective in challenging datasets that do not contain any semantic labels. Our method is the first to model the style densely in these datasets. In the first, third, fourth, fifth and sixth rows, sky in our translations reflect the exemplar style of sky more accurately. In the second row, the yellow leaf in the exemplar stylizes the vegetation (grass) whereas sky is less affected by the style of the yellow leaf.

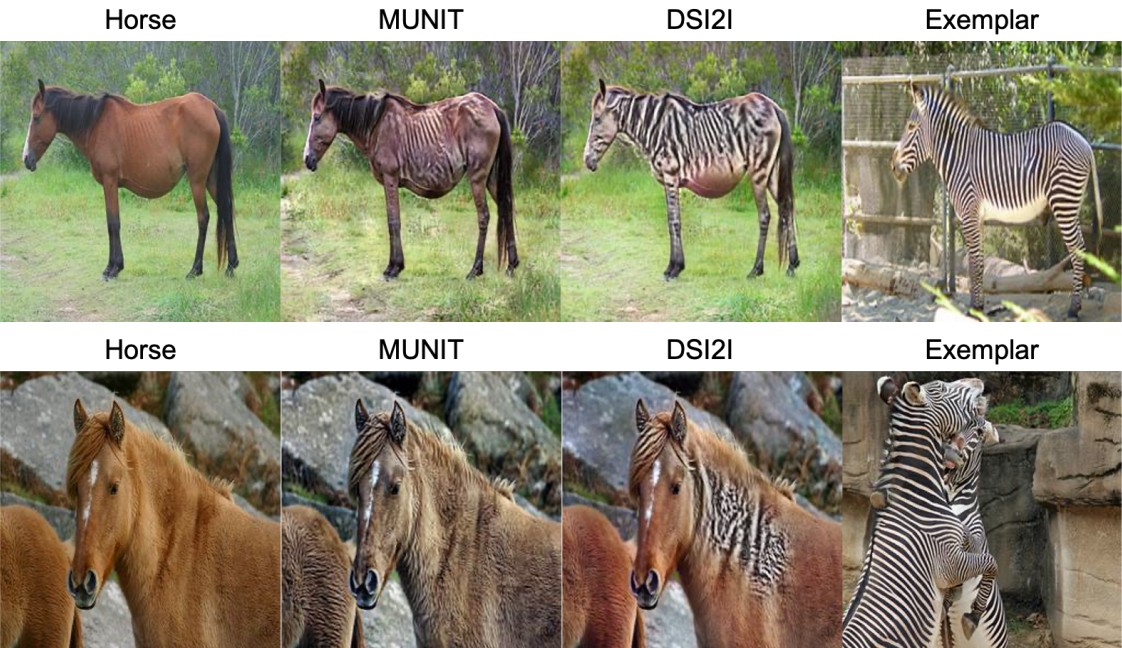

Figure 8: **Limitation on I2I tasks that require semantic changes.** On the horze2zebra dataset that requires semantic changes, our method fails to make the required changes. We present a cherry picked result on the first row. The quality of the outputs of our method is reflected more accurately in the second row where the stripes are not added properly.

## C    Technical Details

In this section, we describe the technical details of the I2I methods that we used in our comparisons. For each method, we adopt the default hyperparameters of the method. All the models are trained for 800K iterations with 224x224 images. We use linear learning rate decay after 400K iterations as suggested in these works.

For training, we resize the input images to have the shorter side of size 256 without changing the aspect ratio and then crop a random 224x224 region. At test time, we generate translations without cropping the images. We use the evaluation code of FSeSim Zheng et al. (2021) for computing the FID. We resize the images to have a shorter side of size 299 without changing the aspect ratio when computing the FID. We borrow the code for IS/CIS from MUNIT Huang et al. (2018). We report the exponential of IS and CIS as done in Huang et al. (2018). The images are resized to have the shorter side of size 299 followed by taking a center crop of size 299x299 as done in Huang et al. (2018). In FID, IS and CIS computations, we sample 100 random source images and 19 target images for each source image. We generate 1900 exemplar based translations as done in Huang et al. (2018).

We use two pre-trained DRN models Yu et al. (2017) for segmentation. We use the pre-trained models for GTA and CS from Hoffman et al. (2018) and Yu et al. (2017), respectively. The former is a DRN-C 26 model whereas the latter is a DRN-D 22.

## D    Results

We provide additional results in this section on CS → GTA. Our method outperforms the baselines in CS → GTA.

| CS → GTA | car | sky | vege-tation | buil-ding | side-walk | road | Avg |
|---|---|---|---|---|---|---|---|
| MUNIT Huang et al. (2018) | 0.57 | 0.44 | 0.59 | 0.53 | **0.47** | 0.35 | 0.49 |
| DRIT Lee et al. (2018) | 0.66 | 0.63 | 0.53 | 0.56 | 0.51 | 0.39 | 0.55 |
| CUT Park et al. (2020) | 0.88 | 0.58 | 0.75 | 0.67 | 0.72 | 0.74 | 0.72 |
| FSeSimZheng et al. (2021) | 0.77 | 0.68 | 0.75 | 0.75 | 0.70 | 0.63 | 0.71 |
| MGUIT Jeong et al. (2021) | 0.76 | 0.69 | 0.69 | 0.65 | 0.68 | 0.57 | 0.67 |
| **DSI2I** | **0.35** | **0.17** | **0.44** | **0.41** | 0.50 | **0.29** | **0.36** |

Table 10: Classwise stylistic distance

| | CS → GTA | |
|---|---|---|
| | FID ↓ | Seg Acc ↑ |
| MUNIT Huang et al. (2018) | 48.91 | 0.79 |
| DRIT Lee et al. (2018) | 48.18 | 0.72 |
| CUT Park et al. (2020) | 65.68 | 0.61 |
| FSeSim Zheng et al. (2021) | 64.81 | 0.74 |
| MGUIT Jeong et al. (2021) | 55.72 | 0.68 |
| **DSI2I** | **45.12** | **0.81** |

Table 11: Fidelity and diversity of the translations. Our method outperforms all others on all metrics.

# E   Semantic Correspondence

## E.1   Marginal Distributions in Optimal Transport

As mentioned in line 497 in the main paper, we discuss a better choice for the marginal distributions for Sinkhorn's Algorithm Cuturi (2013). The most straightforward choice for transportation masses $\mathbf{p}_x$ and $\mathbf{p}_y$ is the uniform distribution. However, doing so transports equal mass from every location in the images. This is problematic for us because we can see in Fig. 1 that when translation pairs have unbalanced classes, the largest semantic region can dominate the style representation and lead to undesired artifacts. In our example in Fig. 1, the content image expects to receive style vectors for roads, buildings, and tree but the exemplar image provides style for sky and road. This results in building and tree regions being stylized by sky attributes.

To solve the unbalanced class problem, we first assume that segmentation labels $\mathbf{M}_x \, \mathbf{M}_y \in \{0,1\}^{K \times HW}$ for $K$ classes are available. We define $\mathbf{M}^k$ as the binary mask for the $k$-th class. The number of pixels in class $k$ is defined as $\tilde{M}^k := \sum_l M^{k,l}$ where $l$ indexes the spatial dimension. We, then, define $\hat{\mathbf{M}}_{yy}, \hat{\mathbf{M}}_{yx} \in \mathbb{R}^{K \times HW}$ as

$$\hat{\mathbf{M}}_{yy} = \mathbf{M}_y^T \tilde{\mathbf{M}}_y \ \text{ and } \ \hat{\mathbf{M}}_{yx} = \mathbf{M}_y^T \tilde{\mathbf{M}}_x \tag{17}$$

where $\tilde{\mathbf{M}} \in \mathbb{R}^{K \times 1}$ is the concatenation of $\tilde{\mathbf{M}}^k$. We propose dividing the transportation mass $\mathbf{p_y}$ of each semantic region in $\mathbf{y}$ by the area of that semantic region to normalize the style based on class distribution of $\mathbf{y}$. We also multiply the mass of each semantic region in $\mathbf{y}$ by the area of the same semantic class in $\mathbf{x}$ to match the expectations of $\mathbf{x}$. We set $\mathbf{p_x}$ to be the uniform distribution and compute

$$\hat{\mathbf{p}}_y = \hat{\mathbf{M}}_{yx} \oslash \hat{\mathbf{M}}_{yy} \tag{18}$$

where $\oslash$ denotes Hadamard (element-wise) division. However, we perform correspondence only during test time and we cannot rely on labels. Hence, we do not know the area of any of the classes. To that end, we propose estimating $\hat{\mathbf{M}}_{yy}$ and $\hat{\mathbf{M}}_{yx}$ based on features $\mathbf{F}_x$ and $\mathbf{F}_y$. As such, we define $\mathbf{Z}_{xx}$ as self-similarity of $\mathbf{x}$ similarly to $\mathbf{Z}_{yx}$ and estimate $\hat{\mathbf{M}}_{yy}$ and $\hat{\mathbf{M}}_{yx}$ with $\mathbf{R}_{yy}$ and $\mathbf{R}_{yx}$ respectively.

$$\mathbf{R}_{yy} = \sum_l \mathbf{Z}_{yy}^l \ \text{ and } \ \mathbf{R}_{yx} = \sum_l \mathbf{Z}_{yx}^l \tag{19}$$

| Source image | Exemplar image | Cosine Similarity |
|---|---|---|

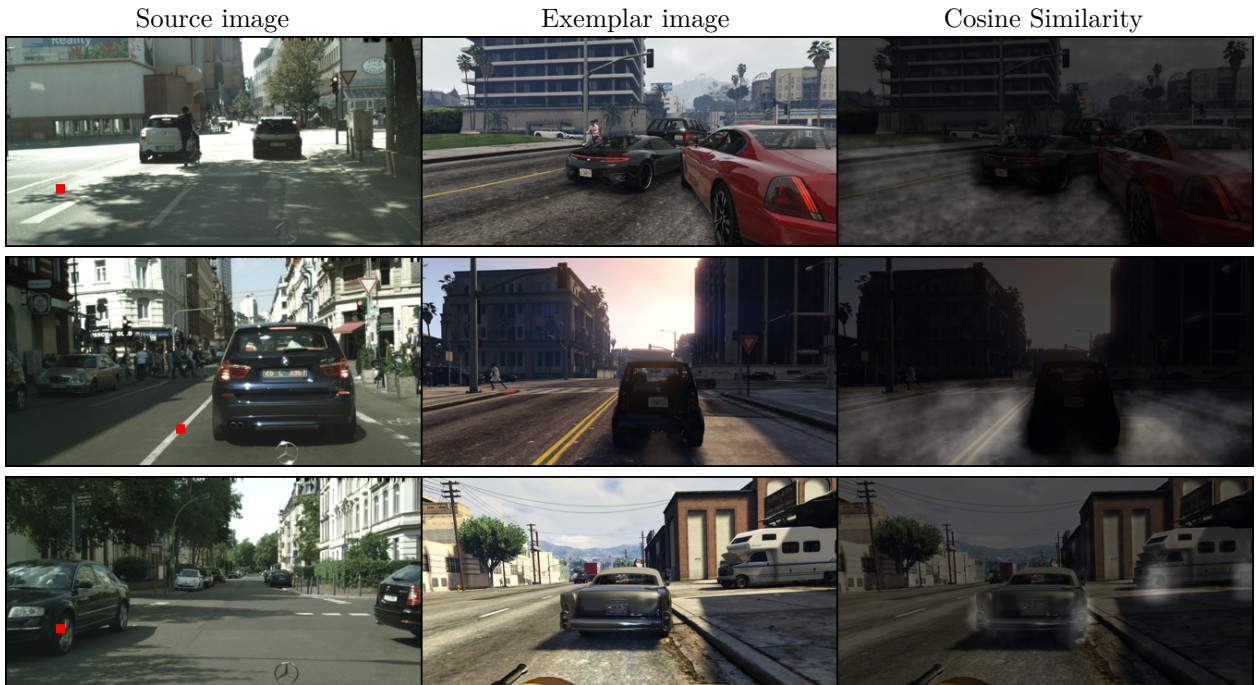

Figure 9: **Visualization of Cosine Similarity across domains.** We choose a region centered at the red point from the source image in the first column and display the cosine similarity between the chosen source region with all the other target regions. Our correspondence module is able to relate the object parts that are not labeled in semantic segmentation annotations which is demonstrated by the correspondence of roadlines in the second row and wheel in the third row.

where $\mathbf{Z}^l \in \mathbb{R}^{HW \times 1}$ and $l$ indexes to the second dimension of $\mathbf{Z}$. We then compute $\hat{\mathbf{p}}_y$ as

$$\hat{\mathbf{p}}_y = \frac{\mathbf{R}_{yx}}{\mathbf{R}_{yy}} \tag{20}$$

which is linearly scaled to obtain a probability distribution $\hat{\mathbf{p}}_y$. Lastly, we compute $\mathbf{A}_{yx}$ as to warp $\mathbf{S}_y^{dense}$ as

$$\mathbf{S}_{y \rightarrow x} = Reshape(\mathbf{S}_y^{dense} \mathbf{A}_{yx}) \; . \tag{21}$$

| Corr Acc | GTA → CS | CS → GTA |
|---|---|---|
| Ours | 0.59 | 0.59 |
| Ours w/o $\mathbf{p}_y$ | 0.57 | 0.56 |

Table 12: Accuracy of semantic correspondence. Our unsupervised $\mathbf{p}_y$ increases the accuracy of correspondence.

### E.2 Effect of the Backbones

We use CLIP Radford et al. (2021) to build semantic correspondences between the two images. CLIP Radford et al. (2021) is trained with image-caption pairs from the internet, to match the global representation of the image with the language representation of the corresponding caption. Hence, it has never received pixel-level supervision or segmentation masks.

We also experiment with a pre-trained DenseCL Wang et al. (2021) model. DenseCL Wang et al. (2021) is trained in a self supervised way to predict the intersection of two crops from two augmentations of the same image. We measure the accuracy of correspondence by warping the segmentation labels of the exemplar via $\mathbf{M}_y\mathbf{A_{yx}}$ and then dividing the correctly classified pixels by the total number of pixels. We observe that semantic correspondence with DenseCL Wang et al. (2021) is less accurate, hence we stick to using CLIP.

We use pre-trained weights for the ResNet50 architecture. Specifically, we extract features from the end of the 'layer1' and 'layer3' stages of ResNet50 architecture.

| Corr Acc | GTA $\rightarrow$ CS | CS $\rightarrow$ GTA |
|---|---|---|
| Ours w/ CLIP Radford et al. (2021) | 0.59 | 0.59 |
| Ours w/ DenseCL Wang et al. (2021) | 0.55 | 0.54 |

Table 13: Accuracy of semantic correspondence with different backbones. Our method uses CLIP Radford et al. (2021) unless otherwise mentioned

### E.3   Ablation on Semantic Correspondence

The contributions of OT are analyzed in the supplementary material in Table 12 and the last two rows of Table 14. Table 12 shows that controlling the marginal distributions in OT leads to more accurate semantic correspondences. The last two rows of Table 14 demonstrate that OT contributes to the performance of our I2I method. OT encourages one-to-one matches and increases the accuracy of these matches (correspondence accuracy in Table 12). As a result, OT leads to better transportation of dense style from the exemplar to the target images (Stylistic distance in Table 14, last two rows), more realistic translations with less artifacts (FID score in Table 14, last two rows), and better content preservation (Segmentation Accuracy in Table 14, last two rows). In addition to Table 12, using softmax instead of OT leads to lower correspondence accuracy in both directions (0.57 -> 0.55 and 0.56 -> 0.53, compared to the last row of Table 12), which supports the theoretical advantage of OT experimentally.

## F   Qualitative Results

### F.1   Ablation Study

We show the qualitative effect of $\mathbf{S}^{mix}$ and $\mathbf{S}^{glb}$ in Fig. 10. Without using $\mathbf{S}^{mix}$ or $\mathbf{S}^{glb}$ with perceptual and adversarial losses, the content component encodes less information about the image, which leads to unrealistic translations with the exemplar style.

| $L_{rand}$ | $L_{glb}$ | $L_{mix}$ | OT | Styl. Dis. $\downarrow$ | FID $\downarrow$ | Seg. $\uparrow$ |
|---|---|---|---|---|---|---|
| ✗ | ✗ | ✗ | ✓ | 0.36 | 59.72 | 0.73 |
| ✗ | ✓ | ✓ | ✓ | 0.38 | 52.82 | 0.75 |
| ✓ | ✓ | ✓ | ✓ | 0.32 | 45.12 | 0.81 |
| ✓ | ✓ | ✓ | ✗ | 0.35 | 45.72 | 0.77 |

Table 14: Ablation studies for loss terms and OT.

### F.2   User Study

We conduct a user study on Amazon Mechanical Turk. We use GTA Richter et al. (2016) images that as the exemplars because they have greater variety in style and appearance. Cityscapes Cordts et al. (2016) images are used as source images. We removed samples which do not contain road and sky. We tried to include complex scenes with multiple objects in the exemplar so that it contains style for many classes. We form 90 source-exemplar pairs. The users are shown one exemplar image and translations from six models, ours and five other baselines Huang et al. (2018); Jeong et al. (2021); Lee et al. (2018); Park et al. (2020); Zheng et al. (2021), as shown in Fig. 11. They are asked to choose the image that looks the most similar to

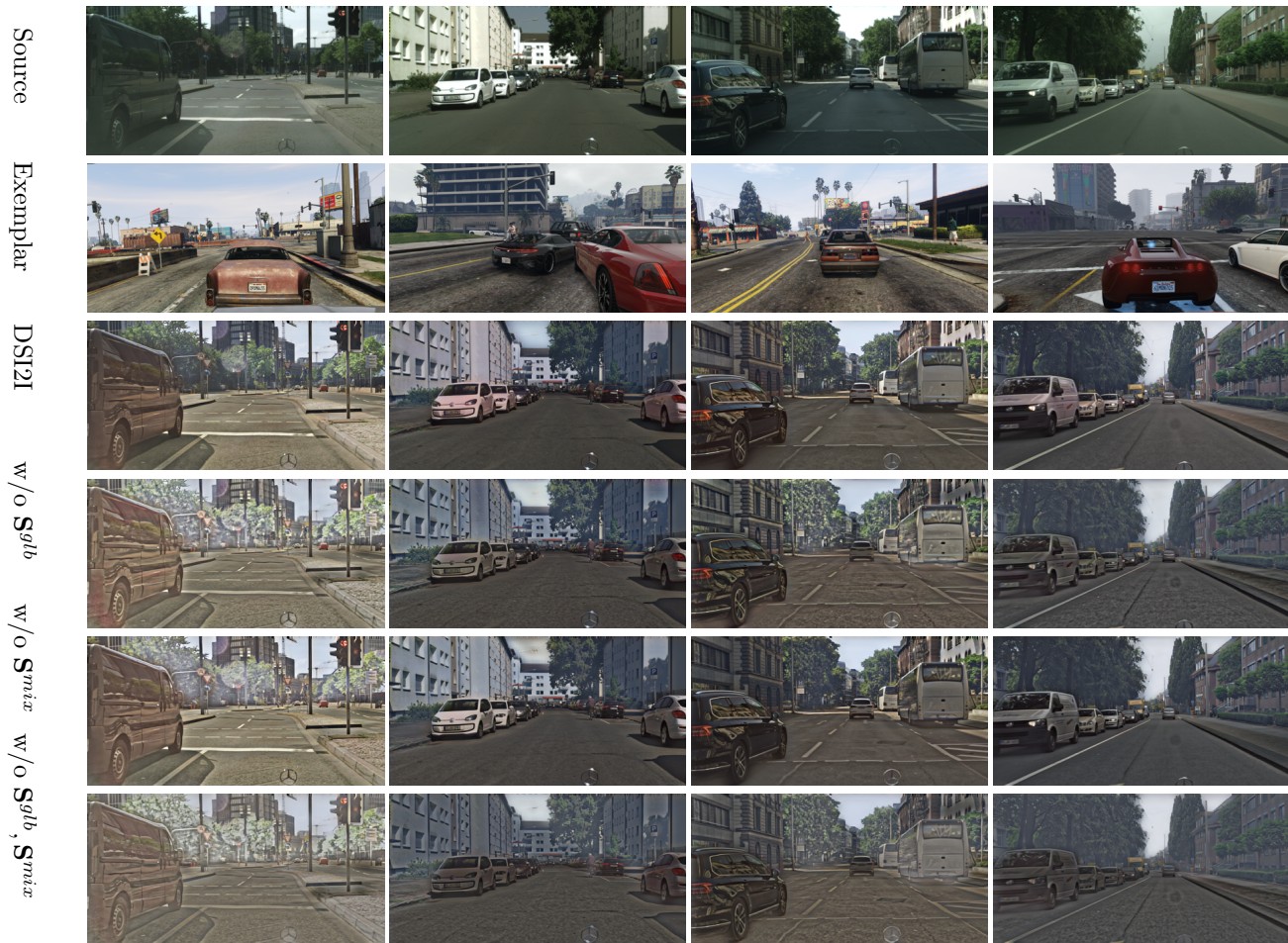

Figure 10: **Effect of $\mathbf{S}^{mix}$ and $\mathbf{S}^{glb}$.** The adversarial and perceptual losses on $\mathbf{S}^{mix}$ and $\mathbf{S}^{glb}$ constrain the dense style representation and, thus, encourage the preservation of content, semantics, and details of the source image. As mentioned in the main paper, the labels are used to swap style across classes in our ablation study instead of the semantic correspondence module. (CS Cordts et al. (2016) to GTA Richter et al. (2016))

the exemplar image. The question they received was: 'Which image is more similar to the target image (T)? Similar images would have closer road and sky colors and would reflect the same time of the day.' We did not provide the users with the source image S because most users were choosing the translations that were closest to the content. We randomized the order of the choices (six methods) in our user study. We also filtered the responses with a mock question in which users are shown the translation of one content image to the style of six exemplars using MUNIT Huang et al. (2018). Only one of the six exemplars is provided in the question as the exemplar. We ask users the same question and accept the answers of those who pick the translation that matches the exemplar displayed in the question. We received answers from 77 users and each user answered 39 questions (+1 mock question). Out of 3003 answers, 1082 picked our method as the best, followed by MUNIT Huang et al. (2018) with 860 votes.

### F.3   Comparison with Other Methods

We provide qualitative examples for our model in the end of our supplementary material.

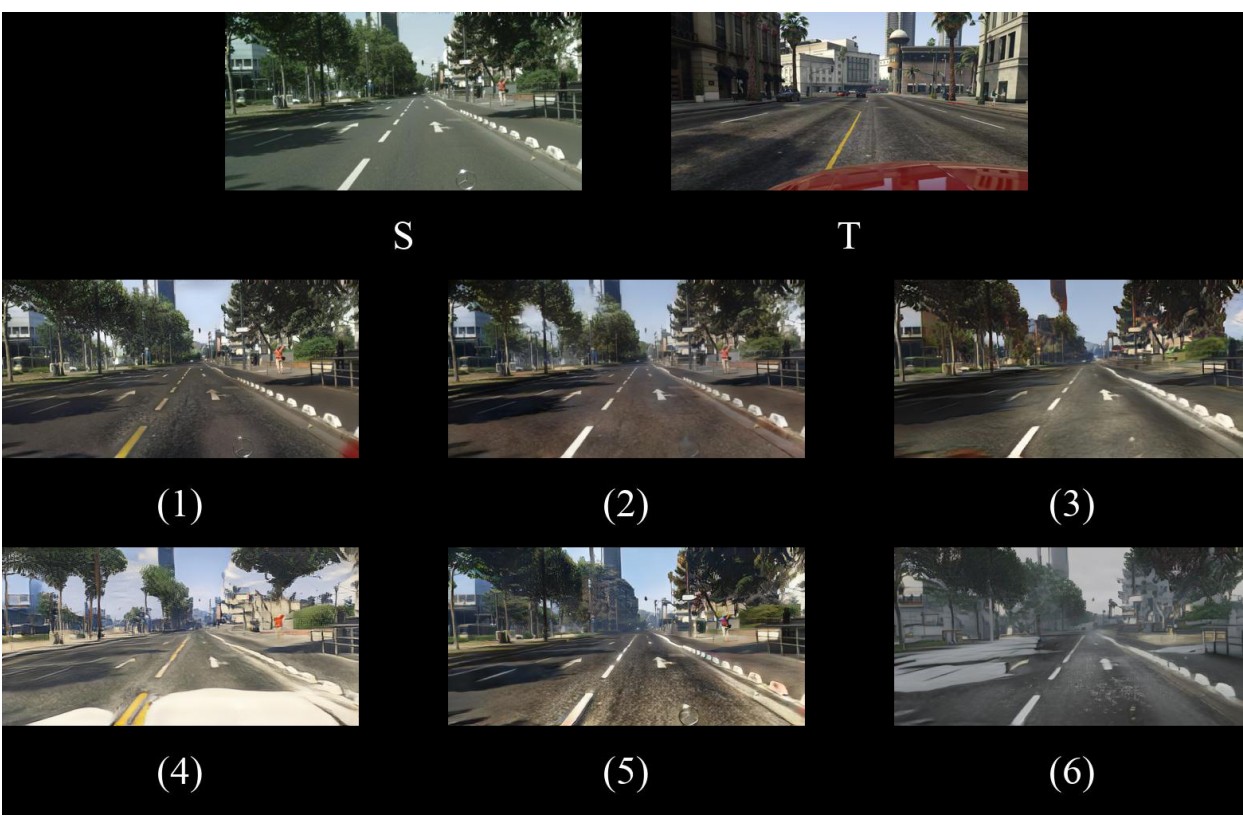

Figure 11: **Screenshot from our user study** The users are asked to pick the translation (from CS Cordts et al. (2016) to GTA Richter et al. (2016)) that looks the most similar to the exemplar image T. We do not provide the users with the source image S and we randomized the order of the choices in our user study. Here, (1): DSI2I, (2): MUNIT Huang et al. (2018), (3): DRIT Lee et al. (2018), (5): CUT Park et al. (2020), (5): FSeSim Zheng et al. (2021), (6): MGUIT Jeong et al. (2021)

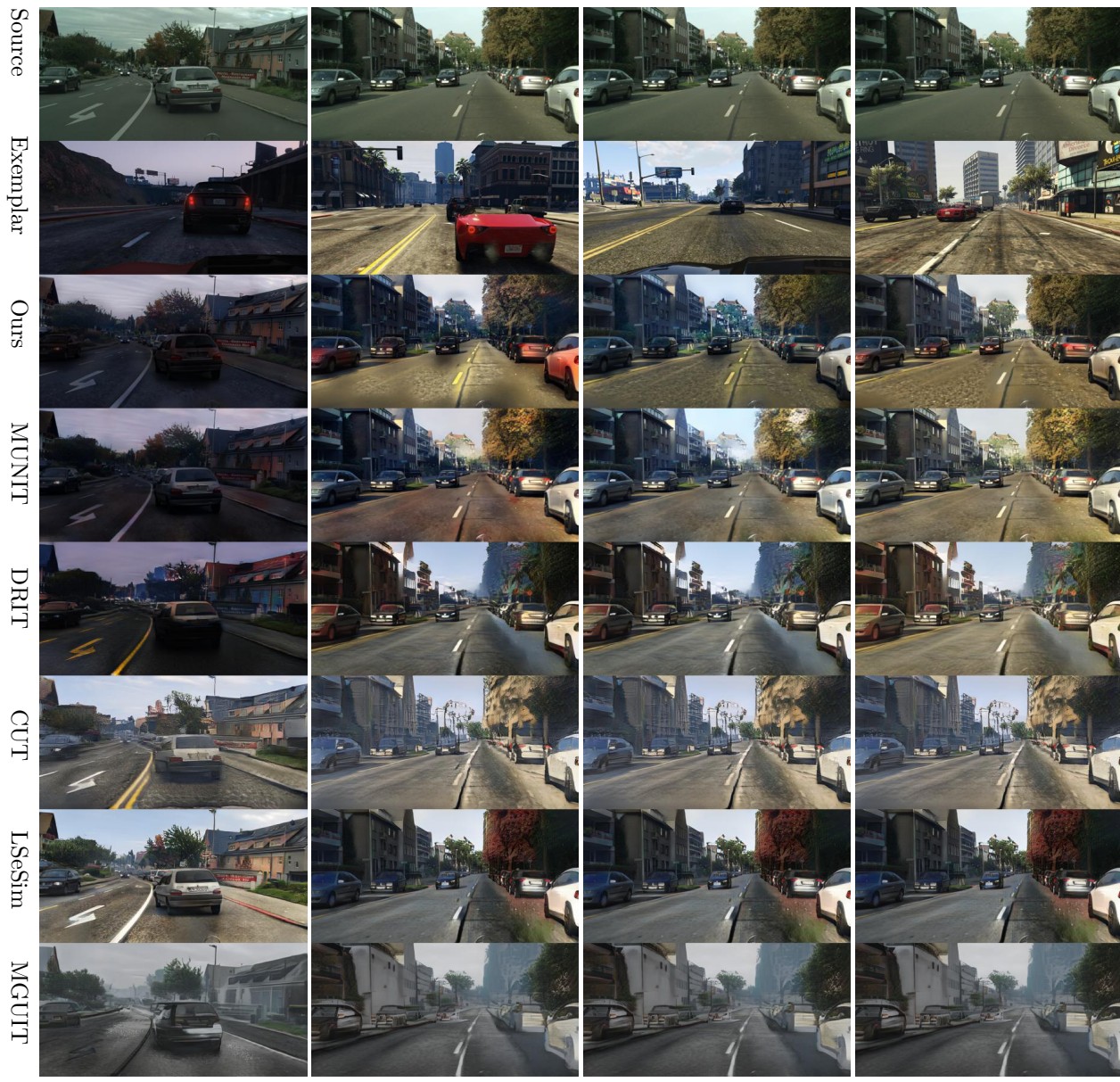

Figure 12: **Qualitative comparison with other methods.** CS → GTA. The road and sky appearance in all the columns are closer to the exemplar road and sky with our method. In the second column, our method is more accurate in the appearance of cars. In the second and third columns, the roadlines are yellow in our translations, which is closer to the exemplar appearance.

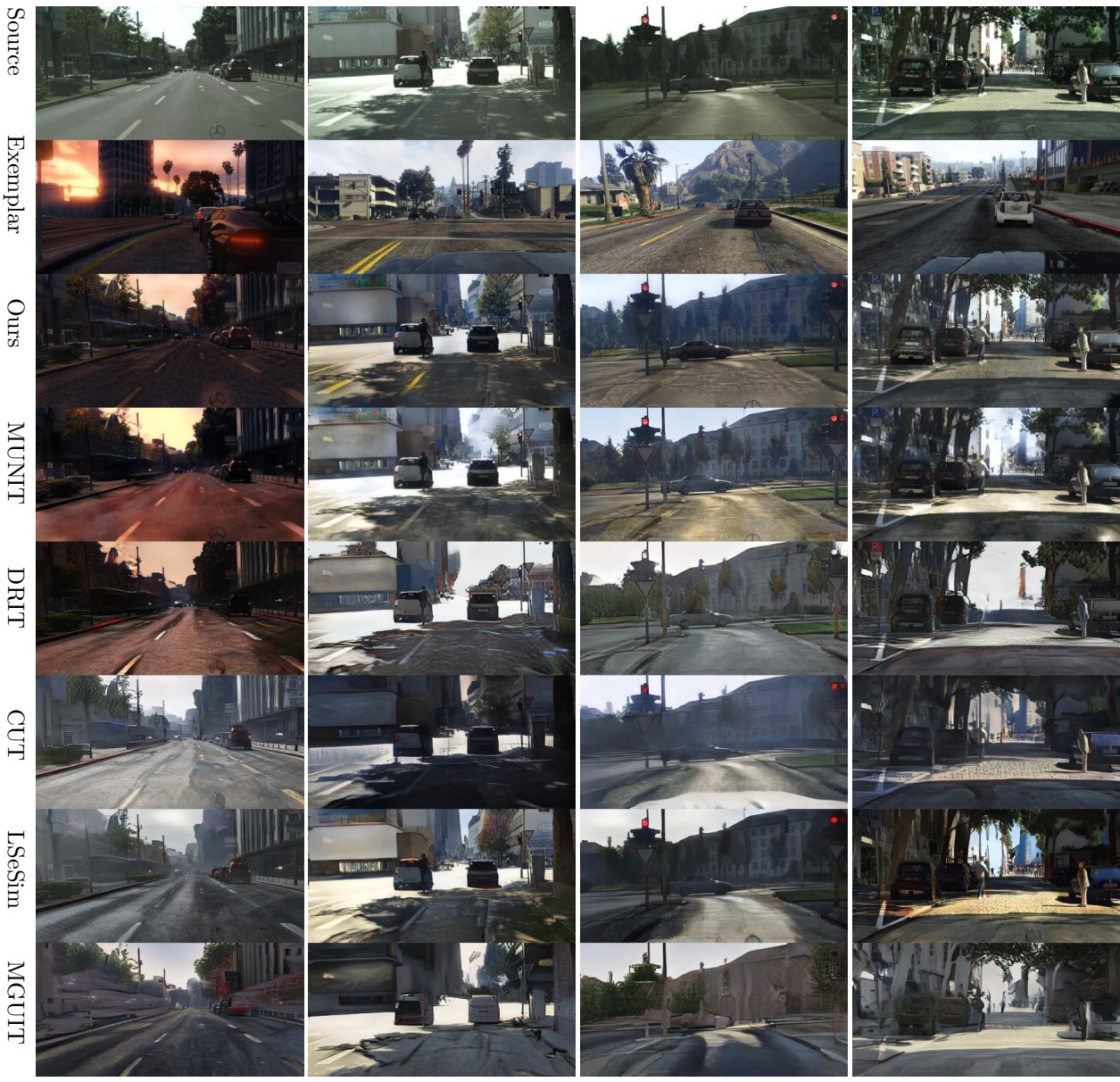

Figure 13: **Qualitative comparison with other methods.** CS → GTA. Our method brings sky and road appearances closer to those of the exemplar in all cases. In the second column, our method preserves the tree whereas the other methods remove it and display sky.

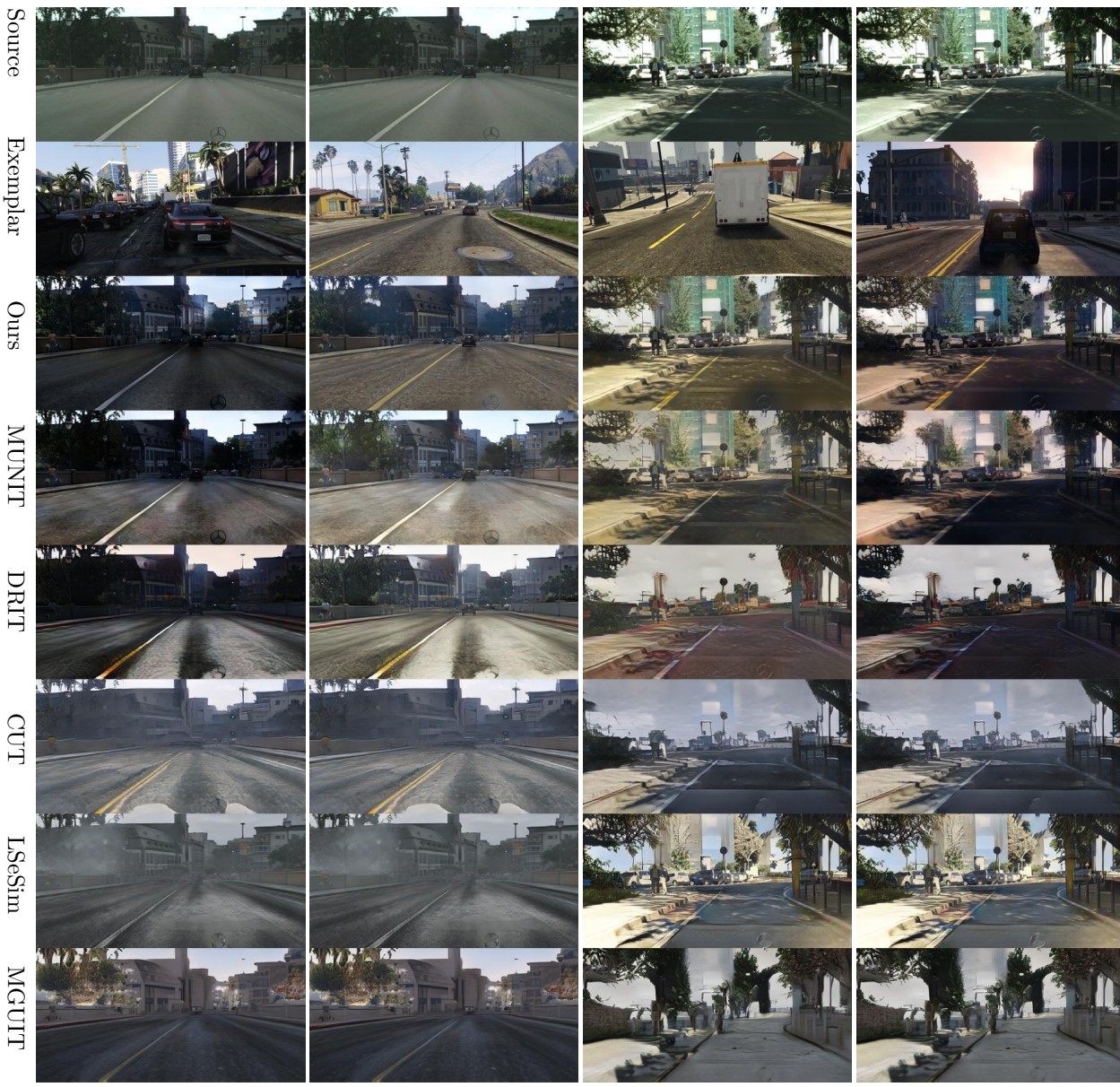

Figure 14: **Qualitative comparison with other methods.** CS → GTA. Our method changes the roadlines based on the exemplar. In the first and second columns, the appearance of the roadlines is adjusted based on the exemplar whereas the other methods either leave them as white or change them to yellow for all the exemplars. In columns three and four, we can see that our method preserves the tree better than the other methods do.

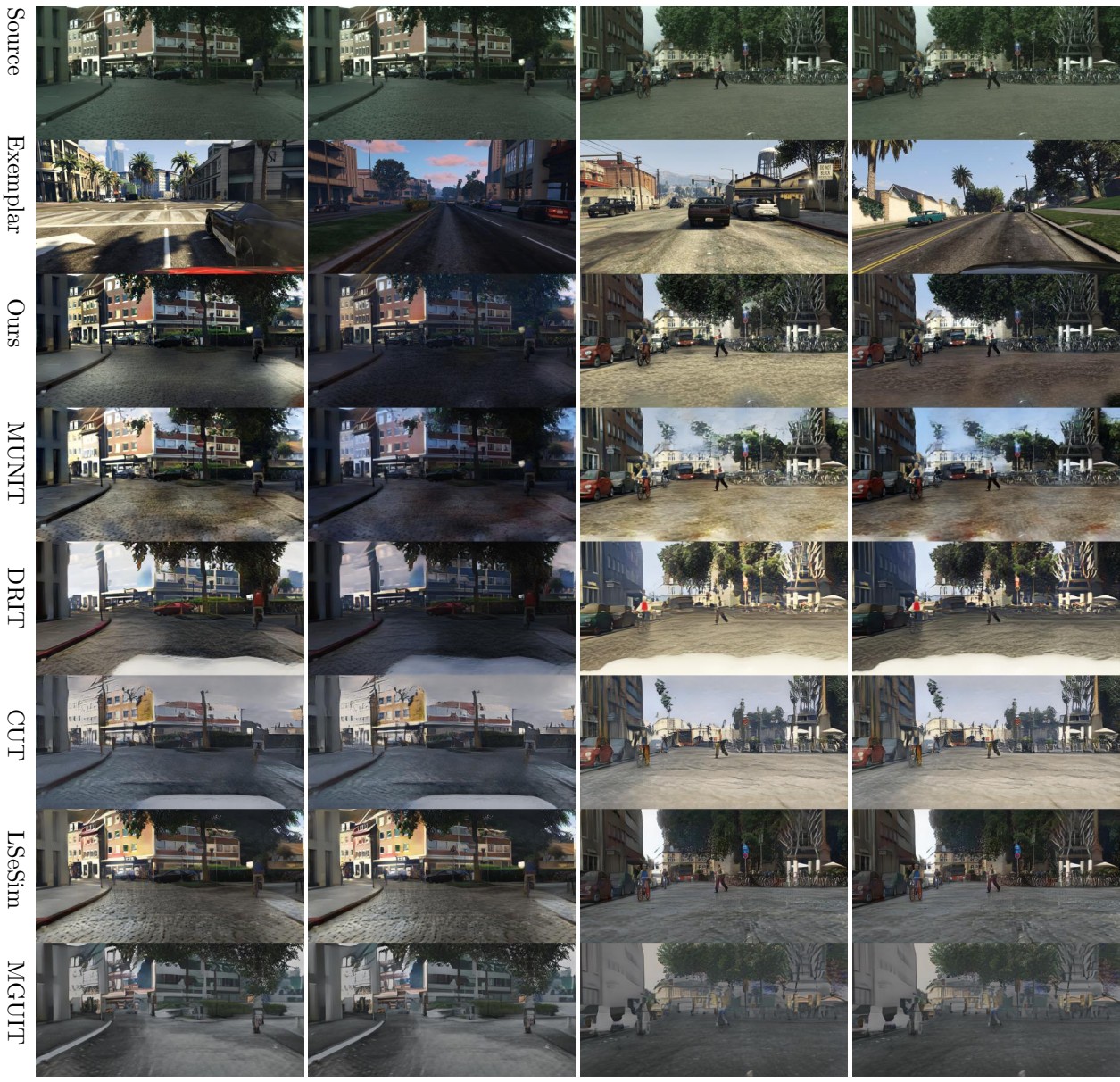

Figure 15: **Qualitative comparison with other methods.** CS → GTA. Our method preserves the building pixels in the first two columns. In the last two columns tree and sky are better preserved with our method and reflect closer appearance to the exemplar.

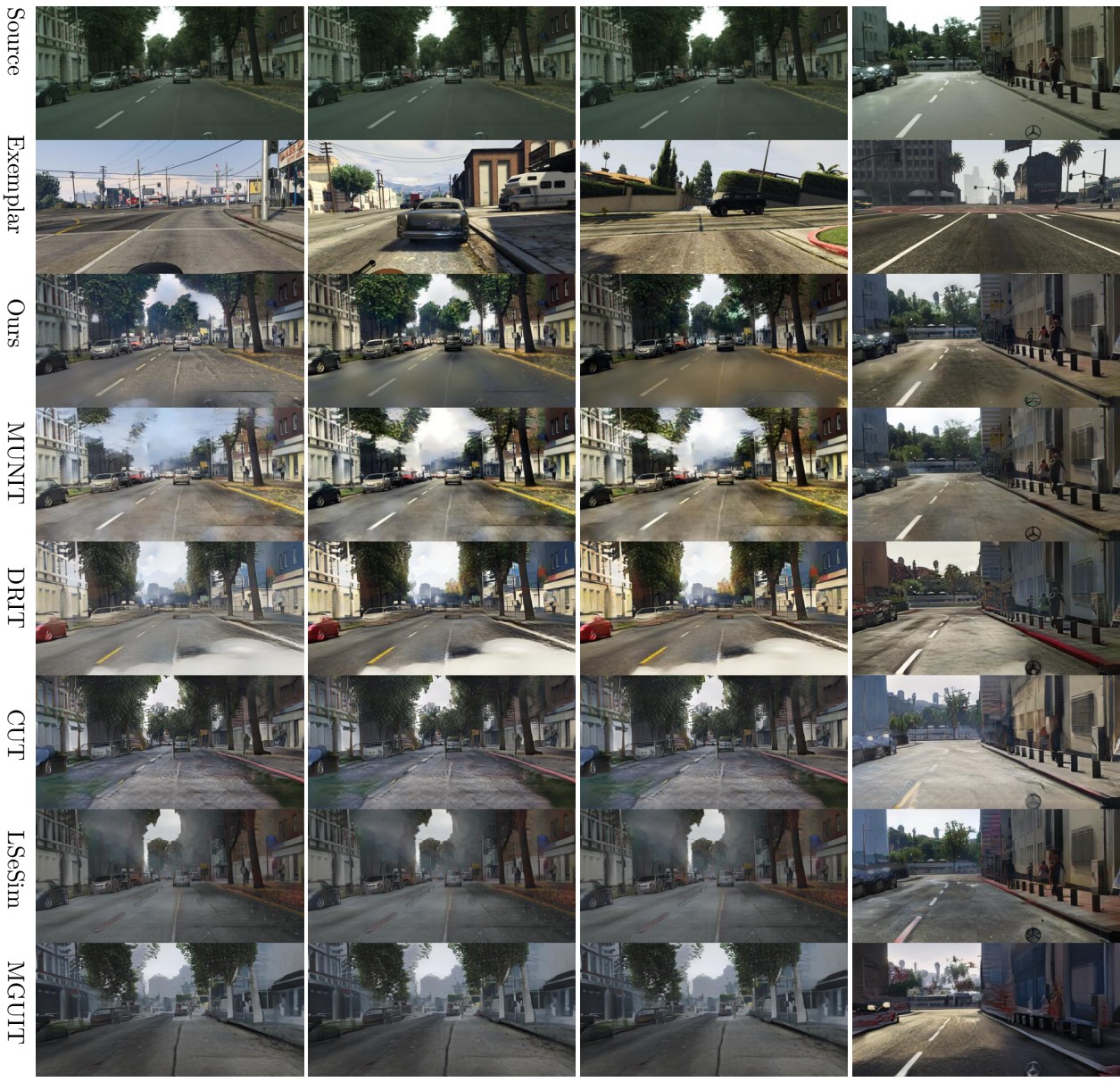

Figure 16: **Qualitative comparison with other methods.** CS → GTA. Our method yields a high output diversity, yet preserves the trees in the first, second, and third columns. The road has the closest appearance to the exemplar with our method in the last column.

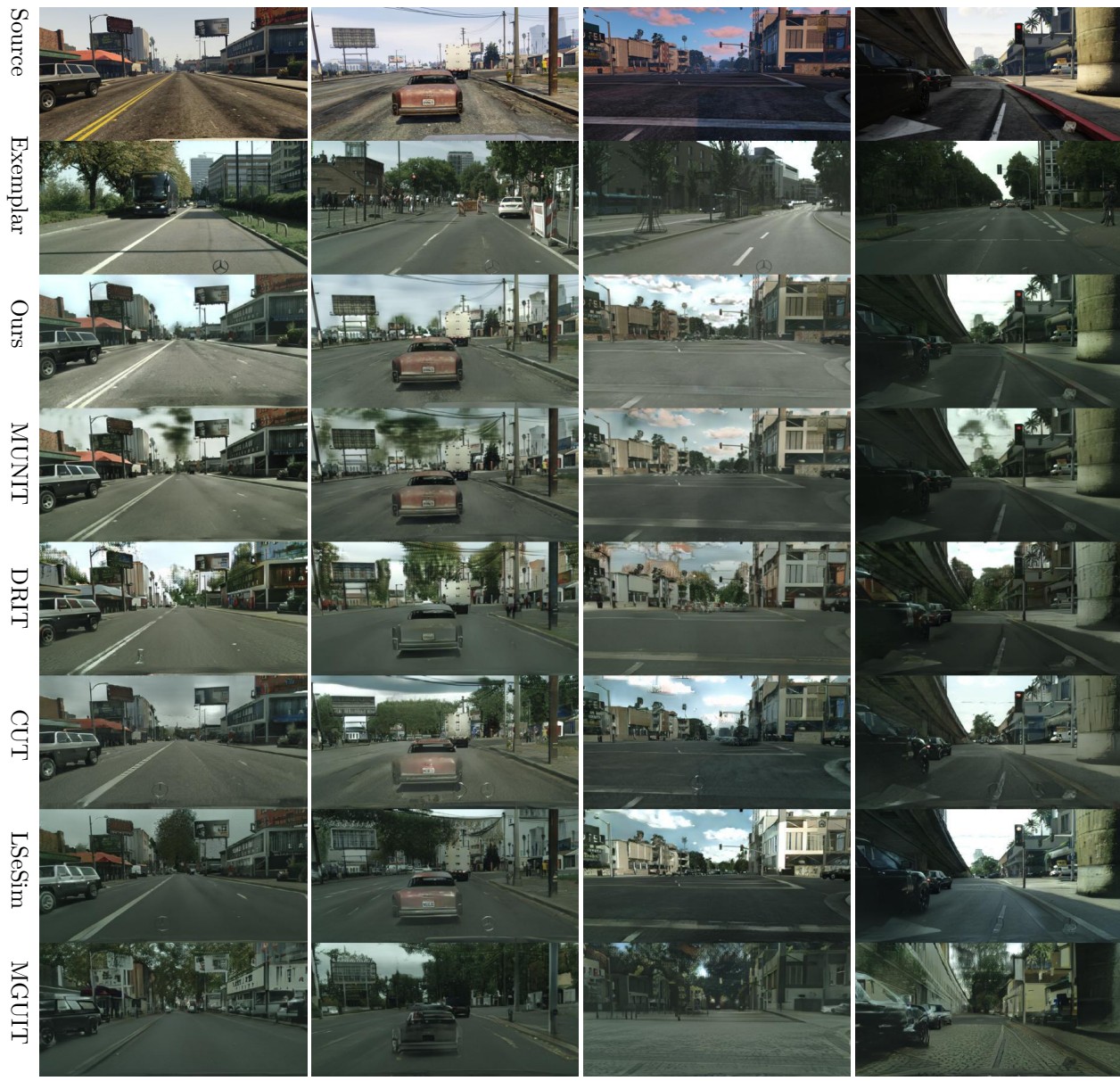

Figure 17: **Qualitative comparison with other methods.** GTA → CS. In this figure, and in the following ones, we show translations in the opposite direction, namely from GTA to CS. Even though stylistic diversity is less in the real image domain, the advantage of our method is still visible. Our method is better at matching the road and sky colors. In the second column, our method does not introduce trees instead of sky, which is common in translations of GTA images.

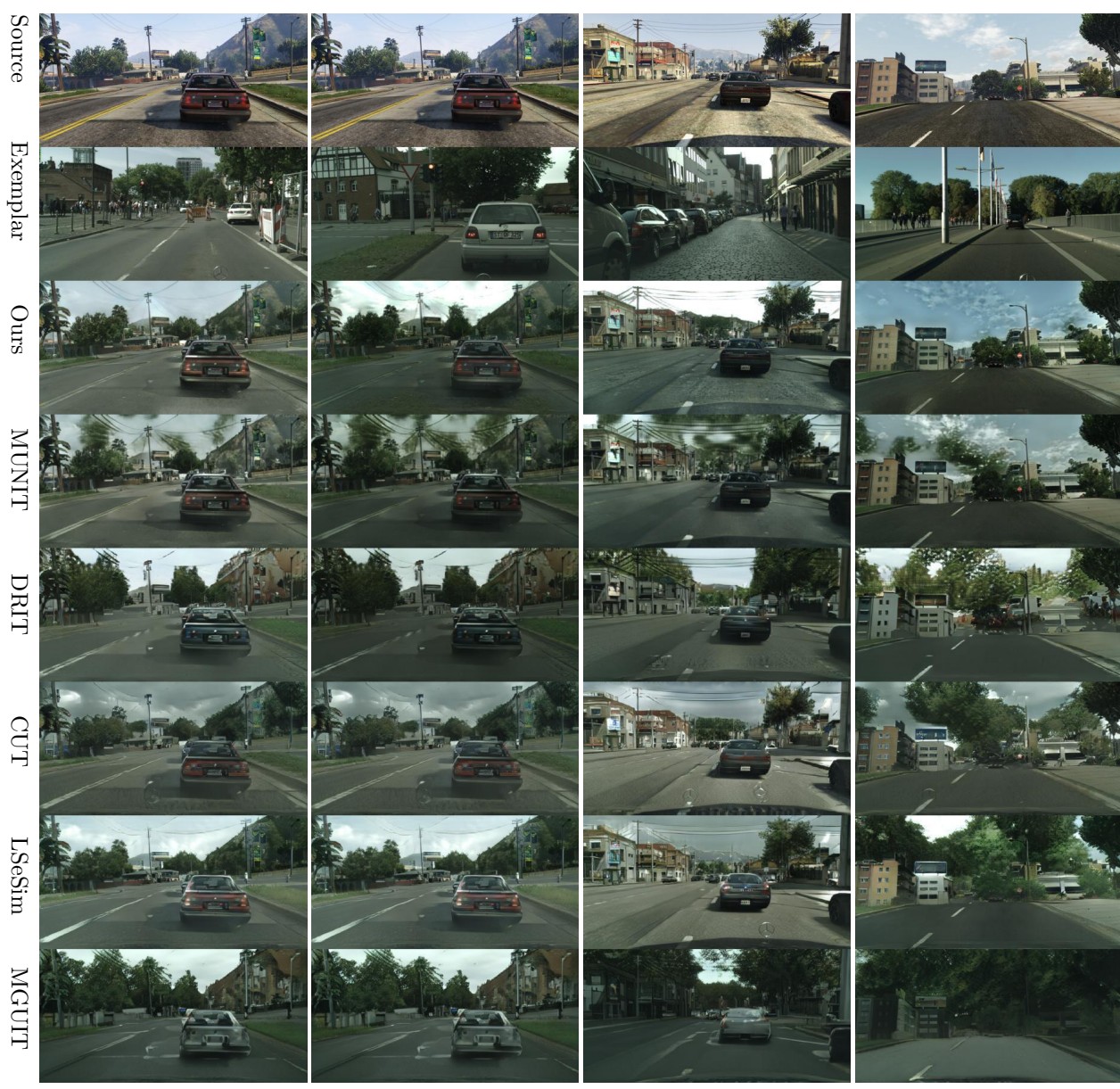

Figure 18: **Qualitative comparison with other methods.** GTA → CS. Aside from road and sky colors, our method is better at preserving the sky regions whereas other methods introduce trees.

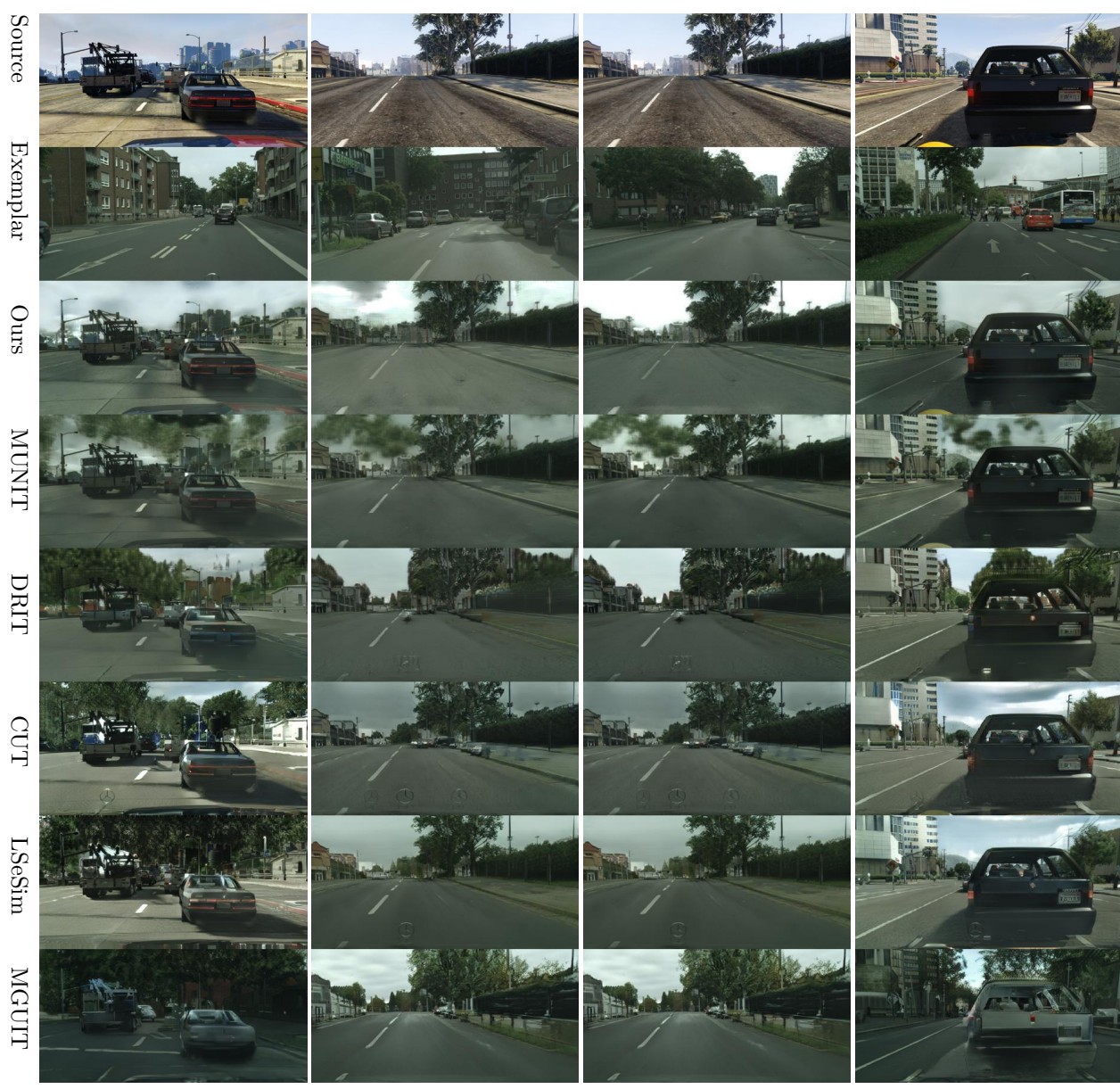

Figure 19: **Qualitative comparison with other methods.** GTA → CS. In all the columns, sky is flipped to tree with other methods. Our method is better at preserving the semantics, yet has diverse outputs.

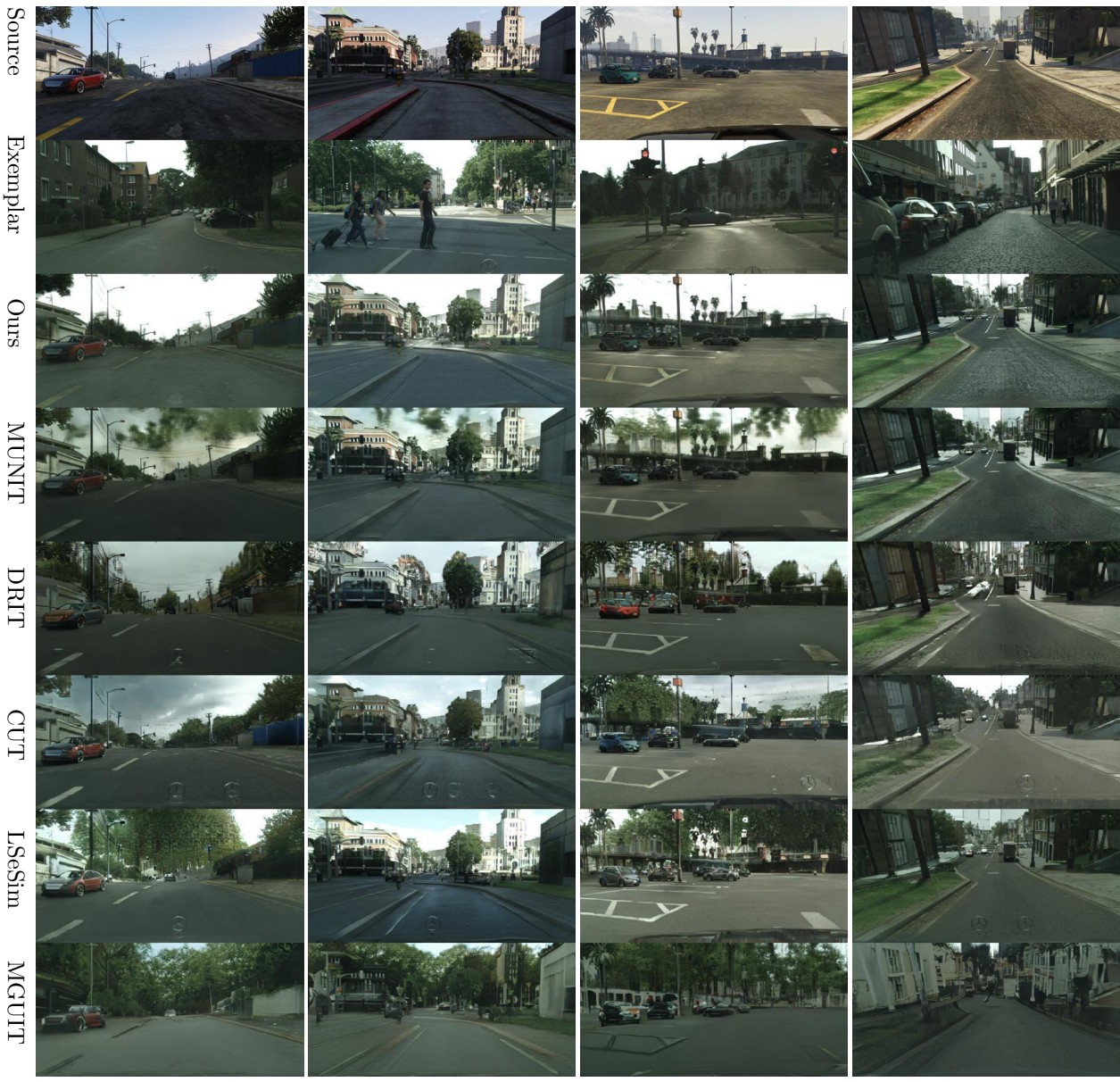

Figure 20: **Qualitative comparison with other methods.** GTA → CS. Our method has much less artifacts in sky in the first three columns. In the last column, the road has closer appearance to the exemplar with our method.

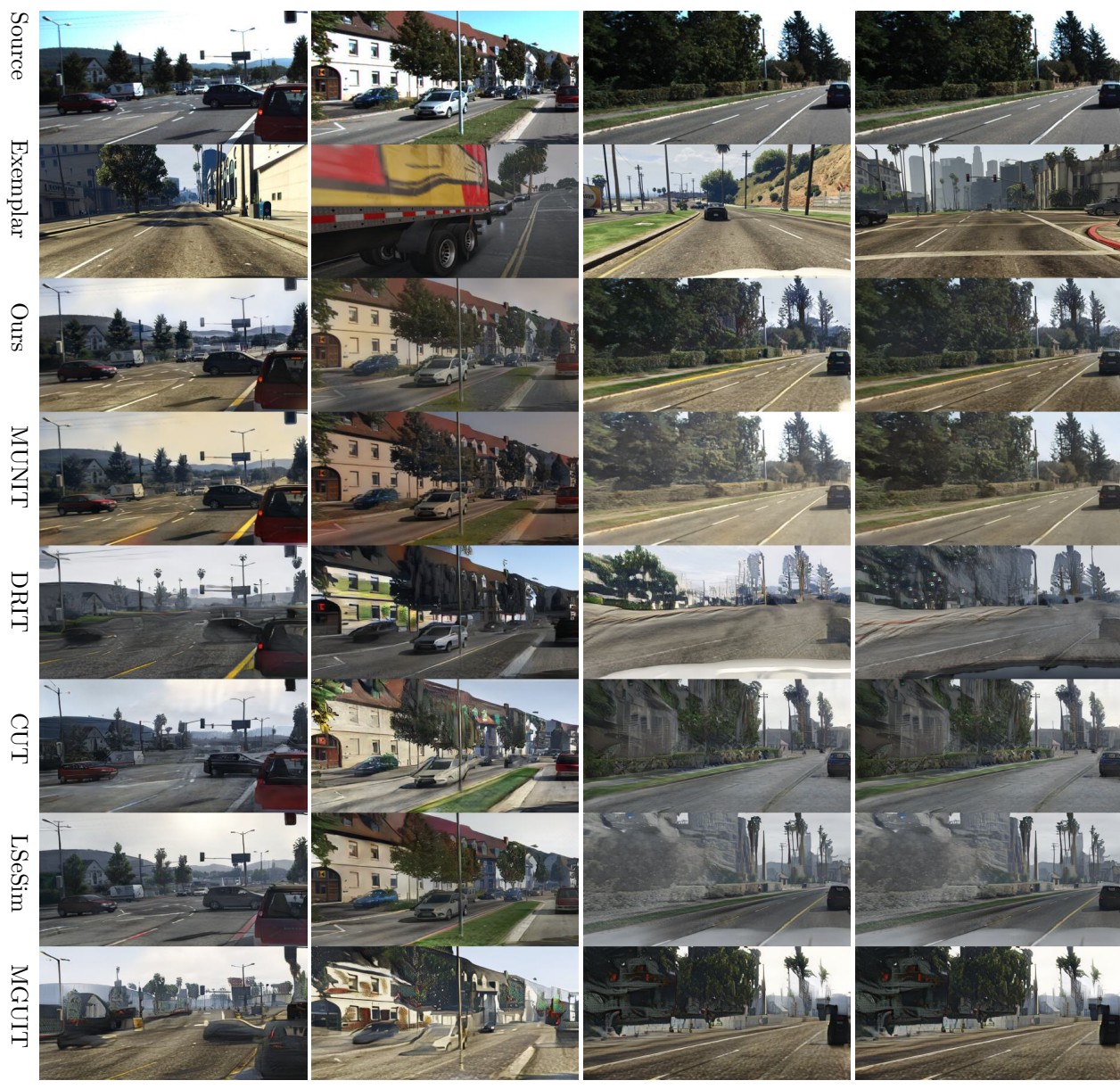

Figure 21: **Qualitative comparison with other methods.** KITTI → GTA. The road and sky appearance in all the columns are closer to the exemplar road and sky with our method. In the second column, red colors from the truck pollute the style of the other areas with MUNIT.

