# OpenReview forum: "DSI2I: Dense Style for Unpaired Exemplar-based Image-to- Image Translation"
_TMLR — Accepted by TMLR_

### Review · Reviewer_wcQB · 2023-11-19

**Summary Of Contributions:**

This work proposes a new method for Unpaired Exemplar based Image-to-Image translation (UE2I2I), replying on dense representations instead of a global feature vectors.

The method use an L1 reconstruction loss to ensure the content (C) and style (S) contain all the information needed to generate an image.
The method uses an adversarial loss and perceptual losses to constraint the behavior of the C repesenations, the global S represnetations and mixed global/dense S_{mix} representations.

To inject a new style into an image given an examplar, the method warmps the style of the target image onto the spatial layout of the source image by solving an optimal transport problem on CLIP dense representations. Given the warp map (which is HWxHW), the method can obtained a warped dense style vector, and feed it into the generators with the the source content vector to get a final generation.

They demonstrate qualitivately and quantitiatively improved results across multiple datasets.

**Audience:**

Yes

**Claims And Evidence:**

Yes

**Requested Changes:**

The work would be strengthened if:
- A figure was added showing the setup if ablations on the method design were worked into the main text of the paper.

**Strengths And Weaknesses:**

*Strengths*
- strong empirical results, in both qualitative and quantitative evaluation
- inclusion of a user study helps ground the results

*Weaknesses*
- The method has many components (3 different adversarial and perceptual losses), separate backbone for style warping.
         - It's hard to follow why it's all necessary and the ablation studies don't have much discussion.

---

> ### Author Response · Authors · 2024-01-12
> **Reasoning for our losses and style components**
>
> Thank you for the constructive comments and your valuable time. We included our reasoning for the losses and style components under our general Official Comments:
>
> - 'Discussion on the ablation studies' (https://openreview.net/forum?id=mrJi5kdKA4&noteId=bFaD1rxZtq) and
>
> - 'Intuition for the style components' (https://openreview.net/forum?id=mrJi5kdKA4&noteId=4Kch1Cmtxx).

---

### Review · Reviewer_wKEx · 2023-11-20

**Summary Of Contributions:**

The paper introduces a novel approach for UEI2I translation using dense style representations. Existing methods in this area represent style using a single global vector per image or per object instance. In contrast, this work proposes modeling style as a dense feature map with spatial dimensions equal to the content representation. This allows for fine-grained control over style transfer to different semantic regions in a complex scene without requiring external semantic supervision.
The method is evaluated on multiple datasets for tasks like synthetic-to-real, real-to-synthetic and real-to-real image translation, showing that the translations are more diverse, better preserve source content, and are closer to exemplar styles compared to state-of-the-art methods​​.

**Audience:**

Yes

**Broader Impact Concerns:**

I have no broader impact concern.

**Claims And Evidence:**

Yes

**Requested Changes:**

Expanding the range of examined styles, especially those with distinct or challenging characteristics, would provide a more comprehensive assessment of the model's versatility.  This could involve testing the method on painting to photograph, summer to winner, label to photo, and their respective reversals.

**Strengths And Weaknesses:**

**Strengths**:
- The idea of dense style representation and transferring style locally between semantic regions is novel. Additionally, the method's ability to establish style correspondences without the need for explicit semantic labels or segmentation masks during training is a commendable advancement, showcasing a pioneering approach in fully unsupervised style disentanglement.
- The proposed method is technically sound with appropriate loss formulations and architectures. Experiments follow standard protocols and compare fairly to state-of-the-art approaches. The metrics cover different desirable properties.
- Modeling style densely could be impactful for controllable image synthesis and editing applications.

**Weaknesses**
- The method relies on off-the-shelf components like CLIP features and optimal transport for correspondence, however the interplay with the proposed components is not extensively analyzed.
- The experiments, though extensive in demonstrating content preservation, appear to focus on a somewhat limited range of styles, which may question the generalizability of the work.
- The examples presented in Figure 9 implies a possible issue in the disentanglement of style and content within the model. Specifically, in row3, column 2/4, cars in these translated image in variably adopt the color of the car in the exemplar, regardless of their original color. This potentially indicates overlap or confusion between style and content elements. This observation raises concerns about the model’s ability to distinctly separate stylistic attributes (like color or texture) from the inherent content features (like the shape or type of objects).

---

> ### Author Response · Authors · 2024-01-12
> **Analysis of the proposed componenets OT, CLIP and the style components**
>
> Thank you for the constructive comments and your valuable time. We included the analysis of our model components (OT, CLIP, losses and the style components) under our general Official Comments:
>
> - 'Discussion on the ablation studies' (https://openreview.net/forum?id=mrJi5kdKA4&noteId=bFaD1rxZtq) and
> - 'Intuition for the style components' (https://openreview.net/forum?id=mrJi5kdKA4&noteId=4Kch1Cmtxx).

---

> ### Author Response · Authors · 2024-01-12
> **Generalizability of the work**
>
> We discuss the applicability of our method to other challenging unlabeled datasets (summer to winter and painting to photograph) under our general Official Comments
>
> - 'Results on other datasets' (https://openreview.net/forum?id=mrJi5kdKA4&noteId=cbUKQbJoHF) and
> - Figure 6 in the revised version

---

> ### Author Response · Authors · 2024-01-12
> **Discussion on disentanglement**
>
> Our goal is to exchange style across semantically related regions. We define the attributes that vary within the domain but do not distort semantics and domain fidelity as the style component. Within this definition, color is part of the dense style and can be exchanged across semantically related regions. Hence, we see the third row, second/forth columns in Figure 9 as successful exchange of dense style as the color of the road lines and the cars are exchanged accurately. We believe the shapes/classes of the objects are well separated from color or texture in our work (supported by Semantic Accuracy and Stylistic Distance metrics).
>
> Note that for the cases where one prefers to preserve the original colors, instead of exchanging colors based on semantic correspondence,  our method can be used to exchange the global styles. Global style translation is possible thanks to the losses on S^glb, which are borrowed from MUNIT. The performance of our method in global style transfer is very similar to that of MUNIT.

---

### Review · Reviewer_Lmht · 2023-12-30

**Summary Of Contributions:**

This work presents a novel approach for exemplar-based image-to-image translation with state-of-the-art performance called DSI2I. The key idea behind DSI2I is modeling the style of the exemplar image using a dense feature grid, that then gets warped to the the content of the source image based on optimal transport. This allows for content to be preserved while making style transfer is more accurate. Disentangled representations through style and content encoder are enabled by perceptual and adversarial losses. Quantitative evaluation is performed on KITTI, GTA and CityScapes, and a user study further confirms the improved quality of DSI2I compared with baselines.

**Audience:**

Yes

**Broader Impact Concerns:**

There are no ethical concerns.

**Claims And Evidence:**

Yes

**Requested Changes:**

- Revise/clarify the writing in the method section, particularly the training portion of figure 2. There are 7 loss terms in total being defined, and understanding them entirely from the text is difficult.
- Expand on possible limitations of the approach where the semantics of the exemplar and source image may be significantly different.
- Fix the vertical spacing of labels in figure 4

**Strengths And Weaknesses:**

Strengths
- Improved quantitative performance relative to prior work across multiple metrics, including the proposed classwise stylistic distance. Qualitatively it is easy to observe the improved precision in the local style transfer as opposed to how prior work appears to globally (and inaccurately) alter all colors of an image. Figure 4 is a great example.
- The approach is well motivated: it makes sense to model exemplar style using dense features rather than single global features to preserve source content and achieve high style fidelity. The results show that this is effective.
- The experiments, analysis and ablation study are very thorough. Additional details in the appendix should aid with reproducibility if code is not released.

Weaknesses
- The writing in the method section can be improved (the training diagram of figure 2 can be improved so that it's easier to follow from the text)
- While for autonomous driving-like domains where there are many shared semantics the proposed idea makes a lot of sense, but would this still be a viable strategy using other types of data as exemplars (for example artist paintings)? It would be good to expand on the limitations of dense modeling, perhaps relating to generalization across domains that may be very different semantically. This is relevant for broader applicability to creative applications.

---

> ### Author Response · Authors · 2024-01-12
> **Revising the intuition for the model components and losses**
>
> Thank you for the constructive comments and your valuable time. We added a discussion about the intuition behind our design choices to the revised version of our paper (at the end of the method section). Details can also be found under our general Official Comments:
>
> - 'Discussion on the ablation studies' (https://openreview.net/forum?id=mrJi5kdKA4&noteId=bFaD1rxZtq) and
> - 'Intuition for the style components' (https://openreview.net/forum?id=mrJi5kdKA4&noteId=4Kch1Cmtxx).

---

> ### Author Response · Authors · 2024-01-12
> **Significantly different source and exemplars**
>
> Our method is effective at preserving the content for a semantically distinct image pair from two semantically related datasets (CS and GTA). The second row of Figure 5 is a good example, where our method preserves the content and styles of the pedestrian, bike and rider classes even though there are no such classes in the exemplar. Another example is provided in monet2photo in Figure 6 (revised version), where the yellow leaf stylizes the vegetation in the ground (a semantically relevant but distinct class) but the other semantic classes are less affected and retain their style. However, our method is not effective for translation between semantically distinct dataset pairs. For example, in horse2zebra dataset, where the image translation requires semantic changes, our method has artifacts with the stripes of the animals based on dense style, in Figure 7 (revised version).

---

### Author Response · Authors · 2024-01-12
**Discussion on the ablation studies**

We thank the reviewers for their constructive comments, valuable time and suggestions.

We acknowledge that our discussion of the ablation studies were too brief and thank the reviewers for pointing this out. We now discuss them in more detail as follows.

Ablations on CLIP and OT.
The contributions of OT are analyzed in the supplementary material in Table 12 and the last two rows of Table 14. Table 12 shows that controlling the marginal distributions in OT leads to more accurate semantic correspondences. The last two rows of Table 14 demonstrate that OT contributes to the performance of our I2I method. OT encourages one-to-one matches and increases the accuracy of these matches (correspondence accuracy in Table 12). As a result, OT leads to better transportation of dense style from the exemplar to the target images (Stylistic distance in Table 14, last two rows), more realistic translations with less artifacts (FID score in Table 14, last two rows), and better content preservation (Segmentation Accuracy in Table 14, last two rows). In addition to Table 14, using softmax instead of OT leads to lower correspondence accuracy in both directions (0.57 -> 0.55 and 0.56 -> 0.53, compared to the last row of Table 12), which supports the advantage of OT experimentally.

Ablation on the losses and the style components.
Tables 8 and 9 include ablations on GTA -> CS for our losses and style components (w/o S^glb is equivalent to w/o L_{adv_glb}, L_{perc_glb}; w/o S^{mix} is equivalent to w/o L_{adv_mix}, L_{perc_mix}).
The adversarial losses are mainly helpful for domain fidelity (Table 9, FID column). The perceptual losses are mainly beneficial for content preservation (Table 9, Seg Acc column).
The first three rows in Table 14 also include ablations on two additional translation setups, KITTI -> GTA (Styl. Dis.) and GTA -> CS (FID and Seg.).
Altogether, our ablations show that the adversarial and perceptual losses on S^{glb} and S^{mix} are useful in terms domain fidelity (FID), content preservation (Seg. Acc.), and stylistic accuracy (Styl. Dis.).

We will incorporate this discussion in the paper.

---

### Author Response · Authors · 2024-01-12
**Intuition for the style components**

We propose using semantic correspondence built from CLIP features but using such a correspondence during training is 1) expensive in terms of computation and memory, and 2) noisy as each point corresponds to all the others with some non-negative weight. For example, a self-correspondence matrix (HWxHW) computed with CLIP would have large diagonal entries, positive but smaller off-diagonal entries for related semantic pixel pairs, and zero off-diagonal entries for semantically unrelated pixel pairs. Instead of computing and storing these noisy and costly matrices with CLIP during training, we provide losses with S^{mix} and S^{glb}.

Our intuition for S^{mix} and S^{glb} is that these two style components replace noisy correspondence matrices of CLIP during training.  S^{glb} is used to model cross-correspondences and can be seen as the output of a uniform, constant HWxHW correspondence matrix of 1/HW s (each content pixel corresponding to all the exemplar pixels equally); S^{dense} can be seen as the output of an identity self-correspondence matrix (each pixel corresponding only to itself); and S^{mix} is the output of a noisy self-correspondence matrix with large diagonal entries and uniform non-diagonal entries (each pixel corresponds mainly to itself but also to all the others). These analytical correspondence matrices enable our model to generalize to the HWxHW cross-correspondence matrices of CLIP, without needing to use CLIP during training.

We will incorporate this discussion in the main paper.

---

### Author Response · Authors · 2024-01-12
**Results on other datasets**

We evaluated our method on datasets which have ground truth segmentation labels. The reason behind this is that, even though our method is applicable to datasets without labels, the metrics we care about (Seg. Acc. and Styl. Dist.) rely on ground-truth segmentation labels. KITTI, GTA and Cityscapes satisfy this label requirement for quantitative evaluation. Furthermore, some of the baselines, CoCosNetv2, MCLNet, MATEBIT and MGUIT, strictly rely on semantic segmentation labels for training, which makes them inapplicable to unlabeled datasets.

Our method, however, is applicable to scenes without semantic labels. Hence, as requested by the reviewers, we provide results on the summer2winter and monet2photo datasets, in both directions (note that we did not evaluate on label2photo as semantic labels have no style, making this dataset not match our application scenario). Our qualitative results show that dense modeling of style enables more accurate transportation of style between semantically relevant regions. (see the revised version, Figure 6)

---

### Decision · Action_Editor_tfun · 2024-03-17

**Recommendation:** Accept with minor revision

**Comment:**

The proposed dense style representation and the use of semantic correspondence for fine-grained style control without seem to be technically sound, and experimental results sufficiently demonstrate improved image-to-image translation and benefits from each component of the proposed method. The authors address most of reviewers' concerns and properly revise the paper to incorporate feedbacks. Therefore, I recommend to accept the paper. However, a few things, especially about the explanation of the proposed training losses, need to be improved in the final version.

- Some training losses need to be more explained. For example, in Eq. (2), what is the role of a random style vector and how is the loss related to rich content representation? More detailed descriptions on motivations, intuitions, and differences between Eq. (2) - (7) are necessary.
- The explanation of S^global and S^mix from the perspective of approximated simulation of semantic correspondence is still unclear. More thorough description with some examples would be necessary.
- Why S^mix is used instead of S^dense?
- There is no explanation of the results in Table 3.

**Audience:**

Unpaired exemplar-based image-to-image translation would be an interesting and important topic in computer vision, and the fine-grained style control by the proposed method can receive a lot of interest from the community including TMLR's audience.

**Claims And Evidence:**

This paper proposes fine-grained style transfer by a dense style representation for the task of exemplar-based image-to-image translation without supervised pairs. The proposed training losses to obtain the dense style representation as well as the use of semantic correspondence by optimal transport on CLIP features at test time seem to be technically sound. Empirical validation shows that the improvement of fine-grained style transfer is significantly achieved by the proposed dense style representation and semantic correspondence in terms of both quantitative metrics and qualitative results, and a number of ablation studies demonstrate benefits from each component of the proposed method.